# Two long-axis dimensions of hippocampal-cortical integration support memory function across the adult lifespan

Kristin Nordin[1,2,3]\*, Robin Pedersen[2,4,5], Farshad Falahati[1,3], Jarkko Johansson[4,6], Filip Grill[7], Micael Andersson[4,5], Saana M Korkki[1,3], Lars Bäckman[1,3], Andrew Zalesky[8,9], Anna Rieckmann[4,5,6,10], Lars Nyberg[4,5], Alireza Salami[1,2,3,4,5]

[1]Department of Neurobiology, Care Sciences, and Society, Karolinska Institutet, Solna, Sweden; [2]Wallenberg Centre for Molecular Medicine, Umeå University, Umeå, Sweden; [3]Aging Research Center, Karolinska Institutet and Stockholm University, Solna, Sweden; [4]Umeå Center for Functional Brain Imaging, Umeå University, Umeå, Sweden; [5]Department of Medical and Translational Biology, Umeå University, Umeå, Sweden; [6]Department of Radiation Sciences, Umeå University, Umeå, Sweden; [7]Centre for Cognitive Neuroimaging, Donders Institute for Brain, Cognition and Behaviour, Radboud University, Nijmegen, Netherlands; [8]Department of Biomedical Engineering, the University of Melbourne, Melbourne, Australia; [9]Department of Psychiatry, the University of Melbourne, Melbourne, Australia; [10]Department of Psychology, University of the Bundeswehr Munich, Munich, Germany

**\*For correspondence:** kristin.nordin@ki.se

**Competing interest:** The authors declare that no competing interests exist.

**Preprint posted** 18 March 2024

**Sent for Review** 18 March 2024

**Reviewed preprint posted** 09 May 2024

**Reviewed preprint revised** 13 February 2025

**Version of Record published** 20 March 2025

## eLife Assessment

This **fundamental** work demonstrates the importance of considering overlapping modes of functional organization (i.e. gradients) in the hippocampus, showing associations with aging, dopaminergic receptor distribution and episodic memory. The evidence supporting the conclusions is **convincing**, although not all analyses were performed in a replication sample. The work will be of broad interest to basic and clinical neuroscientists.

**Abstract** The hippocampus is a complex structure critically involved in numerous behavior-regulating systems. In young adults, multiple overlapping spatial modes along its longitudinal and transverse axes describe the organization of its functional integration with neocortex, extending the traditional framework emphasizing functional differences between sharply segregated hippocampal subregions. Yet, it remains unknown whether these modes (i.e. gradients) persist across the adult human lifespan, and relate to memory and molecular markers associated with brain function and cognition. In two independent samples, we demonstrate that the principal anteroposterior and second-order, mid-to-anterior/posterior hippocampal modes of neocortical functional connectivity, representing distinct dimensions of macroscale cortical organization, manifest across the adult lifespan. Specifically, individual differences in topography of the second-order gradient predicted episodic memory and mirrored dopamine D1 receptor distribution, capturing shared functional and molecular organization. Older age was associated with less distinct transitions along gradients (i.e. increased functional homogeneity). Importantly, a youth-like gradient profile predicted preserved episodic memory – emphasizing age-related gradient dedifferentiation as a marker of cognitive decline. Our results underscore a critical role of mapping multidimensional hippocampal organization in understanding the neural circuits that support memory across the adult lifespan.

## Introduction

The hippocampus plays a critical role in human behavior beyond its well-established involvement in memory and spatial navigation (*Laurita and Spreng, 2017*; *Moscovitch et al., 2016*; *Nadel and Peterson, 2013*). Contemporary views hold that its broad involvement in cognition emerges through the combination of its intrinsic circuitry and its widespread neocortical connections – placing it at the interface of multiple behavioral systems (*Moscovitch et al., 2016*; *Ranganath and Ritchey, 2012*). Characterizing organizational principles of hippocampal integration with the larger neocortical landscape is therefore key to our understanding of its contribution to cognition and to the many diseases associated with its dysfunction (*Barnes et al., 2009*; *Campbell and Macqueen, 2004*; *Harrison, 2004*; *Xie et al., 2020*).

Animal models (*Amaral and Witter, 1989*; *Witter and Amaral, 2021*), together with histological and functional descriptions in humans (*Amunts et al., 2005*; *Libby et al., 2012*; *Plachti et al., 2019*), emphasize the hippocampus transverse (mediolateral) and longitudinal (anteroposterior) axes in determining its functional organization, contribution to behavior (*Grady, 2020*; *Poppenk et al., 2013*), and vulnerability to neurological disease (*Lladó et al., 2018*; *Small et al., 2011*). In humans, functional analogues of the hippocampus canonical internal circuitry and its anatomical connections with neocortical areas have successfully been provided by resting-state functional magnetic resonance imaging (fMRI) (*Libby et al., 2012*; *Dalton et al., 2019*), at a coarse scale confirming the anteroposterior differentiation in connectivity observed in the animal literature (*Amaral and Witter, 1989*; *Aggleton, 2012*; *Strange et al., 2014*). Despite the consistency by which this anteroposterior organizational dimension emerges (*Genon et al., 2021*), significant questions however remain regarding its spatial distribution across cortex and its contribution to behavior across the adult human lifespan.

Lack of consensus is especially evident in terms of the hippocampus functional connectivity with the default-mode network (DMN), encompassing core areas of the brain's system for memory-guided behavior (*Ranganath and Ritchey, 2012*; *Spreng et al., 2009*). Several studies primarily attribute integration with the DMN to the posterior hippocampus (*Ranganath and Ritchey, 2012*; *Libby et al., 2012*; *Adnan et al., 2016*; *Poppenk and Moscovitch, 2011*), consistent with its anatomical connections to midline posterior parietal areas (*Aggleton, 2012*). Other sources emphasize the anterior hippocampus as driving connectivity with the DMN (*Blessing et al., 2016*; *Chase et al., 2015*; *Zhong et al., 2019*), on the basis of its anatomical connections with ventromedial prefrontal areas (*Aggleton, 2012*). These inconsistencies ultimately limit our understanding of hippocampal functional specialization, and the impact on cognition of the heterogeneous vulnerability to aging and disease observed along the hippocampus longitudinal axis (*Small et al., 2011*; *Langnes et al., 2020*). We propose that inconsistencies in part stem from overlooking multiple overlapping and complementary functional modes, not discernable through traditional parcellation-based approaches, which assume homogeneous function within distinct, pre-defined, portions of the hippocampus.

Accumulating evidence in young adults indicate that hippocampal neocortical connectivity is indeed fundamentally multidimensional – organized in several overlapping, but distinct, spatial modes (*Genon et al., 2021*; *Vos de Wael et al., 2018*; *Przeździk et al., 2019*; *Katsumi et al., 2023*; *Tian et al., 2020*). Whereas such gradient-based observations describe a principal anteroposterior mode of neocortical connectivity, they also identify orthogonal modes of long-axis and transverse variation in connectivity (*Vos de Wael et al., 2018*; *Przeździk et al., 2019*; *Katsumi et al., 2023*). Importantly, these hippocampal gradients appear to reflect well-known gradients of macroscale brain function (*Katsumi et al., 2023*), which express functional differentiation across distinct cortical hierarchies (*Huntenburg et al., 2018*; *Margulies et al., 2016*). The principal anteroposterior gradient has been linked to functional differentiation along a task-negative/task-positive cortical dimension (*Fox et al., 2005*), separating neural communities involved in the formation of representations from sensory input (e.g. visual, somatosensory, and DMN areas) and those involved in the modulation of these representations (e.g. frontoparietal areas of attention and control) (*Katsumi et al., 2023*; *Margulies et al., 2016*). In parallel, a secondary, nonlinear, long-axis gradient is suggested to correspond to the principal unimodal-transmodal gradient of cortical function (oppositely anchored in associative DMN areas and in unimodal sensory and motor cortices [*Huntenburg et al., 2018*; *Margulies et al., 2016*; *Katsumi et al., 2023*]). However, the biological underpinnings of this secondary gradient remain still unknown, in contrast to the principal gradient, demonstrated also in microstructure (*Adnan et al.,*

2016), gray matter covariance (*Plachti et al., 2019*; *Ge et al., 2019*), and gene expression (*Vogel et al., 2020*).

Differences in the topography (i.e. the spatial layout) of the principal, anteroposterior, hippocampal gradient may predict episodic memory (*Przeździk et al., 2019*), but current findings are restricted to younger age – and to mainly the same sample (i.e. the Human Connectome Project [*Van Essen et al., 2013*; *Vos de Wael et al., 2018*; *Przeździk et al., 2019*; *Katsumi et al., 2023*]). Moreover, it remains unknown in which capacity the secondary long-axis gradient contributes to behavior. Yet, a comprehensive model of the spatial properties of hippocampal functional organization should consider that hippocampal functional alterations occur across the adult lifespan and manifest across the hippocampus in a heterogeneous manner. Specifically, aging has been linked to differential functional isolation of anterior and posterior hippocampal regions from prefrontal areas and large-scale brain networks including the DMN (*Damoiseaux et al., 2016*; *Salami et al., 2016*). Such neocortical disconnection of hippocampal subregions has been linked to dysfunction during memory encoding and retrieval (*Salami et al., 2016*; *Nyberg et al., 2019*; *Salami et al., 2014*), and, in turn, to episodic memory decline (*Salami et al., 2014*). Findings which link hippocampal isolation to increased functional homogeneity within the region suggest that its disconnection from neocortex is driven by the spatial patterns in which Alzheimer's disease (AD) pathology accumulates in cognitively healthy older adults (*Berron et al., 2021*; *Harrison et al., 2019*). Importantly, loss of functional segregation between task-negative and task-positive poles is a functional hallmark of both healthy aging (*Grady et al., 2016*) and AD (*Weiler et al., 2017*) – introducing overall ambiguity as to whether hippocampal gradients established in young adults persist into older age.

Dopamine (DA) is one of the most important modulators of hippocampus-dependent function (*Edelmann and Lessmann, 2018*; *El-Ghundi et al., 2007*) and influences the brain's functional architecture through enhancing specificity of neuronal signaling (*Seamans and Yang, 2004*). Consistently, there is a DA-dependent aspect of maintained functional network segregation in aging which supports cognition (*Pedersen et al., 2023b*). Animal models suggest heterogeneous patterns of DA innervation (*Gasbarri et al., 1994*; *Kempadoo et al., 2016*) and postsynaptic DA receptors (*Dubovyk and Manahan-Vaughan, 2019*), across both transverse and longitudinal hippocampal axes, likely allowing for separation between DA modulation of distinct hippocampus-dependent behaviors (*Edelmann and Lessmann, 2018*). Moreover, the human hippocampus has been linked to distinct DA circuits on the basis of long-axis variation in functional connectivity with midbrain and striatal regions (*Kahn and Shohamy, 2013*; *Nordin et al., 2021*). Taken together with recent findings revealing a unimodal-transmodal organization of the most abundantly expressed DA receptor subtype, D1 (D1DR), across cortex (*Pedersen et al., 2024*), we tested the hypothesis that the organization of hippocampal-neocortical connectivity partly reflects the underlying distribution of hippocampal DA receptors, predicting predominant spatial correspondence for any hippocampal gradient conveying a unimodal-transmodal pattern across cortex.

Here, we characterize the multidimensional functional organization of the hippocampus in two independent adult-lifespan samples, and map individual differences in fine-scale topographic properties of connectivity gradients onto behavioral and molecular phenotypes. We report three hippocampal gradients displaying distinct correspondence to (a) canonical gradients of cortical function, (b) the organization of hippocampal DA receptors, and (c) individual differences in memory function. Multivariate, data-driven, classification on gradient topography identified older adults exhibiting a youth-like gradient profile and superior memory function as distinct from age-matched older counterparts, emphasizing a behavioral significance of preserved functional hippocampal topography in older age.

## Results
### Multiple dimensions of hippocampal-neocortical integration across the adult lifespan

Connectopic mapping (*Haak et al., 2018b*) was applied to resting-state fMRI data (n=180, 90 men/90 women; 20–79 years; mean age = 49.8 ± 17.4) from the DopamiNe, Age, connectoMe, and Cognition (DyNAMiC) study (*Nordin et al., 2022*). For replication, we used an independent sample of 224 adults (122 men/102 women; 29–85 years mean age = 65.0 ± 13.0) from the Betula project

(*Nilsson et al., 2004*). Connectopic mapping was used to extract the dominant modes of functional cortical connectivity within the hippocampus based on nonlinear manifold learning (Laplacian eigenmaps) applied to a similarity matrix derived from connectivity fingerprints computed between each hippocampal voxel and each voxel within neocortex. This identified a set of orthogonal connectopic maps (i.e. eigenvectors) describing overlapping connectivity topographies (i.e. gradients) within the hippocampus. Gradients were computed at subject level and at group level across the sample, separately for the left and right hippocampus. We analyzed the first three gradients, together explaining 63% and 71% of the variance in left and right hemispheres, respectively. This number corresponded to a clear elbow in the scree plot (*Appendix 1—figure 1*).

The principal gradient (G1), explaining 44% and 53% of the variance in left and right hemispheres, was organized along the hippocampus longitudinal axis, conveying gradual anterior-to-posterior variation in cortical connectivity (*Figure 1A*). This pattern of connectivity change is illustrated by dividing subject-level G1 connectopic maps into 23 long-axis bins of ~2 mm and plotting the average gradient values as a function of their distance from the most anterior hippocampal voxel (*Przeździk et al., 2019*; *Figure 1B*). The second-order gradient (G2), explaining 11% of the variance in both hemispheres, expressed a secondary long-axis gradient organized from the middle hippocampus toward anterior and posterior ends (*Figure 1A and B*). Finally, the third-order gradient (G3), explaining 8% and 7% of the variance, reflected variation along the hippocampus transverse axis, such that inferior-lateral parts of the hippocampus were separated from medial-superior parts (*Figure 1A*). This pattern was most pronounced in the anterior hippocampus. Inspecting G3 across sample-specific segmentations of cornu ammonis (CA1–3), dentate gyrus (DG/CA4), and subiculum subfields suggested that while CA1–3 expressed the full extent of the gradient, and DG/CA4 variation around its center, the subiculum expressed only the most inferior section of the gradient (*Appendix 1—figure 2*).

The three gradients reflected gradients identified in young adults (*Vos de Wael et al., 2018*; *Przeździk et al., 2019*; *Katsumi et al., 2023*) and were highly reproducible in the independent replication dataset (*Appendix 1—figure 3*). Correspondence between samples was determined by spatial correlations (left hemisphere: G1: r=0.990, p<0.001; G2: r=0.946, p<0.001; G3: r=0.918, p<0.001; right hemisphere: G1: r=0.996, p<0.001; G2: r=0.969, p<0.001; G3: r=0.897, p<0.001). Furthermore, the reliability of connectopic mapping to produce functional connectivity gradients was determined across varying degrees of spatial smoothing and contrasted against connectopic maps derived from random data (*Watson and Andrews, 2023*). Results confirmed high stability of resting-state gradients and their efficacy in capturing interindividual differences, whereas random data failed to produce meaningful gradients (*Appendix 1—figure 4*).

## Hippocampal gradients reflect distinct dimensions of macroscale cortical organization

The projection of G1 onto cortex conveyed a pattern linking medial orbitofrontal, temporolimbic, and medial parietal regions at the anterior end of the gradient with occipital and frontoparietal regions at the posterior end (*Figure 1C*). For further characterization, we computed G1 gradient values within seven cortical networks (*Yeo et al., 2011*) and examined their position in gradient space. This placed the DMN, limbic, and somatomotor networks at anterior-to-middle parts of the gradient, whereas visual, ventral attention, and frontoparietal networks toward the posterior end of the gradient (*Figure 1D*). In contrast, G2 exhibited a unimodal-transmodal pattern across cortex, linking the middle hippocampus to frontal and posterior parietal regions, and anterior and posterior hippocampal ends to somatomotor and occipital regions (*Figure 1C*). Consistently, the DMN and frontoparietal network mapped onto G2 at one end, and visual and somatomotor networks at the other (*Figure 1D*). Across cortex, G3 primarily separated temporal and insular areas from medial parietal and medial frontal areas (*Figure 1C*). Aligning with cortical connectivity previously reported for hippocampal subfields (*Vos de Wael et al., 2018*; *de Flores et al., 2017*), areas of the DMN most strongly mapped onto the most inferior end of G3, consistent with the connectivity profile of the subiculum, whereas ventral attention and somatomotor networks had a stronger medial position along G3, aligning with reported connectivity of CA1–3 (*Figure 1D*).

Next, we linked the cortical patterns of gradients to the established macroscale layout of cortical function (*Margulies et al., 2016*) using Spearman's rank correlation analyses. G1 showed significant correspondence with the cortical representation-modulation gradient (*Margulies et al., 2016*),

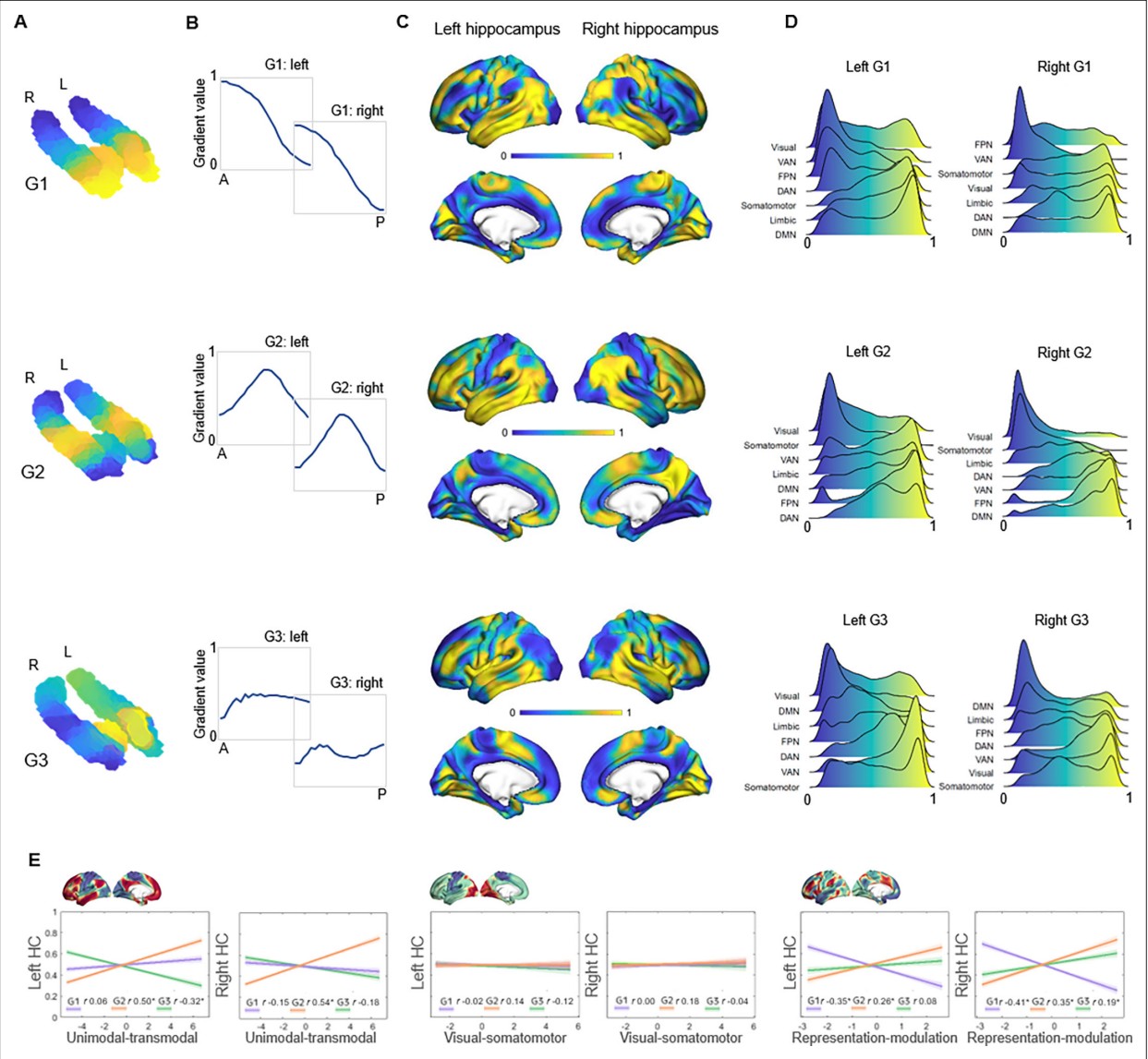

**Figure 1.** Topographic gradients of hippocampal cortical connectivity. (**A**) The first three hippocampal connectopic maps (G1–G3), explaining 67% of the variance across left and right hemispheres. Similar colors convey similar patterns of cortical connectivity. Values range between 0 (blue) and 1 (yellow). (**B**) Plots convey change in connectivity along the anteroposterior hippocampal axis. Mean values from 23 hippocampal bins (each ~2 mm) are plotted against their distance (in mm) from the most anterior hippocampal voxel. Values were estimated based on subject-level gradients and averaged across participants. G1 conveys gradual change in connectivity along an anteroposterior gradient. G2 conveys gradual change in connectivity along a second-order long-axis gradient, separating the middle hippocampus from anterior and posterior ends. G3 conveys close to no change in connectivity along the longitudinal axis, with connectivity change instead organized in a primarily medial-lateral gradient. (**C**) Cortical projections for G1, G2, and G3. Values range between 0 (blue) and 1 (yellow). (**D**) The order of cortical networks in gradient space. Density plots visualize the distribution of gradient values for seven cortical networks (*Yeo et al., 2011*). (**E**) Correlations between cortical patterns of hippocampal gradients and the three primary gradients of cortical functional organization, which are exemplified at the top of each graph (*Margulies et al., 2016*).

differentiating the task-negative DMN and somatomotor networks from task-positive areas of attention and control (left G1: Spearman's r=–0.353, $p_{spin}$<0.001; right G1: Spearman's r=–0.406, $p_{spin}$<0.001; *Figure 1E*). G1's correlation with this cortical gradient was greater than its correlations to other cortical gradients (6.3<Z<9.6, ps<0.001), as well as significantly stronger than correlations observed for G2 and G3 (1.6<Z<6.5, ps<0.05). In contrast, G2 corresponded to the principal unimodal-transmodal gradient of cortical function (left G2: Spearman's r=0.502, $p_{spin}$<0.001; right G2: Spearman's r=0.536, $p_{spin}$<0.001; *Figure 1E*) to a greater extent than G1 and G3 (4.9<Z<10.9, ps<0.001) and in comparison to other cortical gradients (5.2<Z<9.4, ps<0.001). Finally, the cortical pattern of G3 showed overall

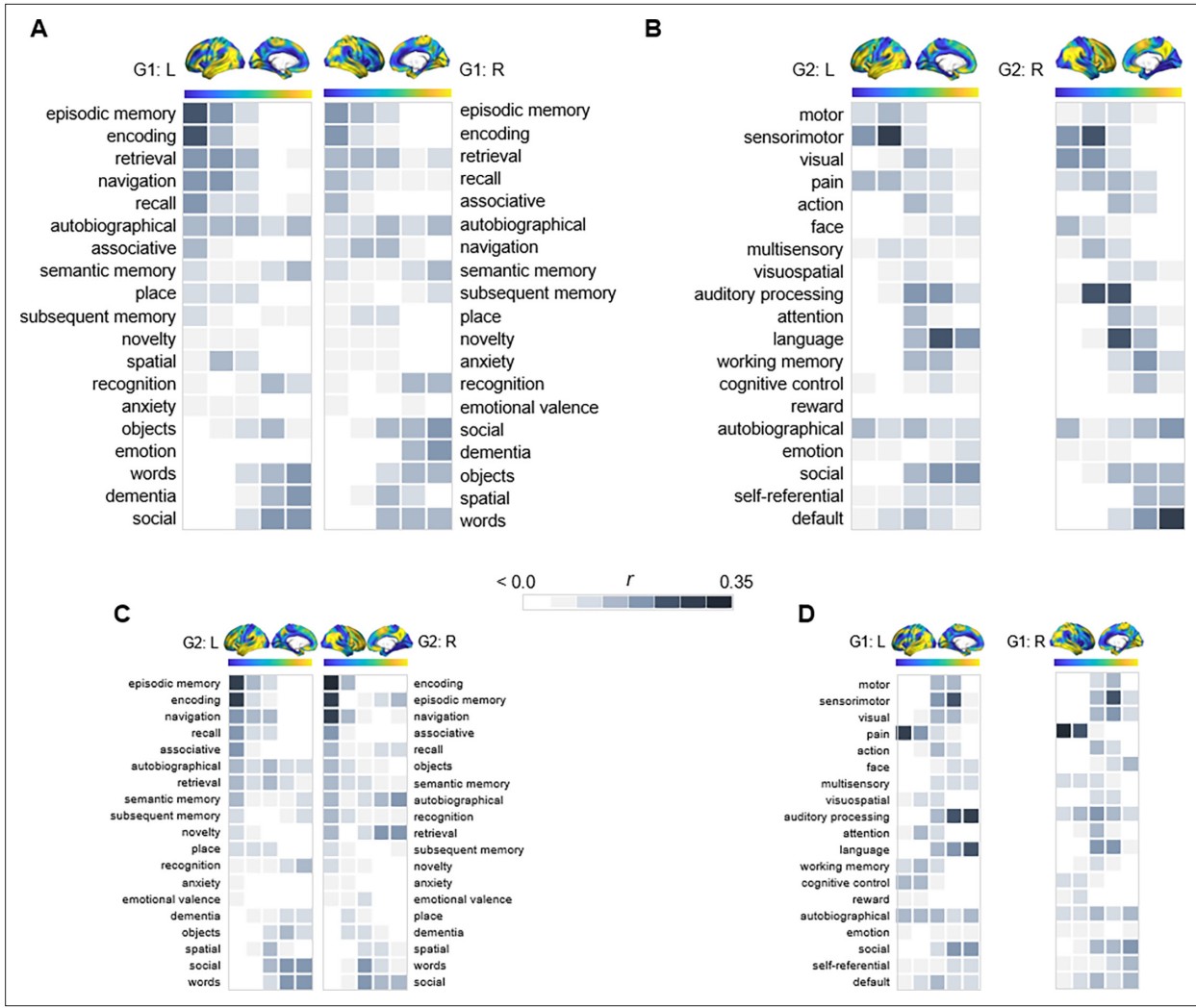

**Figure 2.** Behavioral profiling of G1 and G2 across cortex. For each gradient, columns represent 20 percentile bins of the gradient's cortical projection. Color shadings represent the strength of correlations between gradient bins and meta-analytical maps in Neurosynth. (**A**) Terms commonly linked to anteroposterior hippocampal functional specialization were assessed across G1 and ranked based on their location along the gradient. (**B**) For G2, terms were selected and ordered as to reflect a unimodal-transmodal cortical axis (*Margulies et al., 2016*). (**C**) The correspondence between G2 and behavioral terms commonly linked to anteroposterior hippocampal functional specialization. (**D**) The correspondence between G1 and behavioral terms expressing a unimodal-transmodal axis.

weaker correspondence with the canonical cortical gradients. G3's correlations with the unimodal-transmodal gradient (left G3: Spearman's r=−0.32, $p_{spin}$ = 0.011) and the representation-modulation gradient (right G3: Spearman's r=0.191, $p_{spin}$ = 0.009) were significantly weaker compared with their G2 counterparts ($Z_{(998)}$ = 4.9, p<0.001) and G1 ($Z_{(998)}$ = 5.3, p<0.001). These multiple lines of evidence contribute to a model of hippocampal functional organization across the adult human lifespan in which G1 and G2 constitute local representations of distinct macroscale cortical motifs.

## Distinct patterns of behavioral transitions along G1 and G2

Given the correspondence of G1 and G2 to distinct gradients of cortical function, we characterized their relevance for hippocampal functional specialization by mapping transitions in behavioral domains onto G1 and G2 using meta-analytical decoding in Neurosynth (*Yarkoni et al., 2011*). Correlations were assessed between meta-analytical maps of behavioral terms and 20 percentile bins of each gradient's cortical projection (*Figure 2*). First, a selection of terms commonly linked to anteroposterior hippocampal functional specialization (*Plachti et al., 2019*; *Grady, 2020*) were assessed across G1 and ranked based on their location along the gradient (*Figure 2A*). The strongest anterior loadings on

G1 belonged to terms including *words*, *social*, and *dementia*, whereas terms of *navigation*, *episodic memory*, *encoding*, and *recollection* showed preferential posterior loadings. In contrast, behavioral transitions along G2 were expected to reflect its unimodal-transmodal organization (*Figure 1B*). To this end, terms were selected and ordered based on a previous report demonstrating unimodal-transmodal behavioral transitions across cortex (*Margulies et al., 2016*). G2 expressed a clear separation between *sensorimotor* and *visual* terms at one end from *social, self-referential,* and *default* terms at the other (*Figure 2B*), confirming its unimodal-transmodal organization.

## Topography of G2 reflects distribution of hippocampal DA D1 receptors

Our next aim was to investigate to which extent the distribution of hippocampal DA D1 receptors (D1DRs), measured by [$^{11}$C]SCH23390 positron emission tomography (PET) in the DyNAMiC (*Nordin et al., 2022*) sample, may serve as a molecular correlate of the hippocampus' functional organization. First, to estimate individual differences in gradients' spatial layout, trend surface modeling (TSM) was applied to each subject-level connectopic map (*Przeździk et al., 2019*; *Haak et al., 2018b*; *Haak and Beckmann, 2020*). This spatial statistics approach parameterizes gradients at subject level, yielding a set of model parameters describing the topographic characteristics of each gradient in x, y, z directions (see Materials and methods and *Appendix 1—figure 5* for model selection). Unlike voxel-wise statistical inference on gradients, which overlooks the high interdependence between voxels' gradient values and demands rigorous correction for multiple comparisons, TSM allows for statistical inference across a region's internal heterogeneity using a concise set of independent parameters (*Haak and Beckmann, 2020*). Moreover, by adjusting the number of polynomial terms, TSM facilitates examination of spatial trends across gradients at coarser-to-finer levels (*Haak and Beckmann, 2020*).

Individual maps of D1DR binding potential (BP) were also submitted to TSM, yielding a set of spatial model parameters describing the topographic characteristics of hippocampal D1DR distribution for each participant. D1DR parameters were subsequently used as predictors of gradient parameters in one multivariate GLM per gradient (in total six GLMs, controlled for age, sex, and mean FD). Results are reported with p-values at an uncorrected statistical threshold and p-values after adjustment for multiple comparisons using the Benjamini-Hochberg method to control the false discovery rate (FDR). Individual differences in D1DR topography significantly explained topography of right-hemisphere G2 (F=1.207, p=0.041, $p_{FDR}$ = 0.073; partial $\eta^2$=0.118), but not of G1 nor G3 (F=0.953–1.108, p=0.222–0.596) (*Figure 3*). This association was robust across multiple TSM model orders (*Appendix 1—figure 6*). Complimentary analyses were then conducted to further evaluate G2 as a dopaminergic hippocampal mode by utilizing additional DA markers at group level. First, a map of D1DR distribution was formed by averaging the [$^{11}$C]SCH23390 BP images across DyNAMiC participants (n=176), and a map of D2DR distribution was formed by averaging [$^{11}$C]raclopride BP images in a subsample of DyNAMiC participants (n=20). Previously published maps of DAT (*Dukart et al., 2018*) and FDOPA (https://www.nitrc.org/projects/spmtemplates) were also analyzed. Correlations across group-level TSM parameters (*Oldehinkel et al., 2022*) revealed significant positive associations between G2 and D1, DAT, and FDOPA (D1: r=0.501, p<0.01, $p_{FDR}$ = 0.021; DAT: r=0.378, p<0.01, $p_{FDR}$ = 0.021; FDOPA: r=0.584, p<0.01, $p_{FDR}$ = 0.021; *Figure 3C*), although not D2 (D2: r=0.131, p=0.440, $p_{FDR}$ = 0.528), whereas correlations were not significant for G1 or G3, indicating that G2 best captures shared principles of functional and molecular organization.

## Dedifferentiated gradient topography in older age

Effects of age on gradient topography were assessed using multivariate GLMs including age as the predictor and gradient TSM parameters as dependent variables (controlling for sex and mean framewise displacement [FD]). One model was fitted per gradient and hemisphere, each model including all TSM parameters belonging to a gradient (in total, six GLMs). There was a significant effect of age on topographic characteristics of all three gradients. G1 displayed the greatest effect of age (left: $F_{(9,150)}$ = 5.853, p<0.001, $p_{FDR}$ = 0.003, partial $\eta^2$=0.260; right: $F_{(9,150)}$ = 6.971, p<0.001, $p_{FDR}$ = 0.003, partial $\eta^2$=0.298), followed by G2 (left: $F_{(12,147)}$ = 2.583, p=0.004, $p_{FDR}$ = 0.01, partial $\eta^2$=0.174; right: $F_{(12,145)}$ = 2.635, p=0.003, $p_{FDR}$ = 0.008, partial $\eta^2$=0.179), and G3 (left: $F_{(12,147)}$ = 1.973, p=0.030, $p_{FDR}$ = 0.056, partial $\eta^2$=0.139; right: $F_{(12,145)}$ = 2.082, p=0.021, $p_{FDR}$ = 0.042, partial $\eta^2$=0.147). To visualize effects, subject-level G1 and G2 values were plotted along the anteroposterior axis, averaged within young

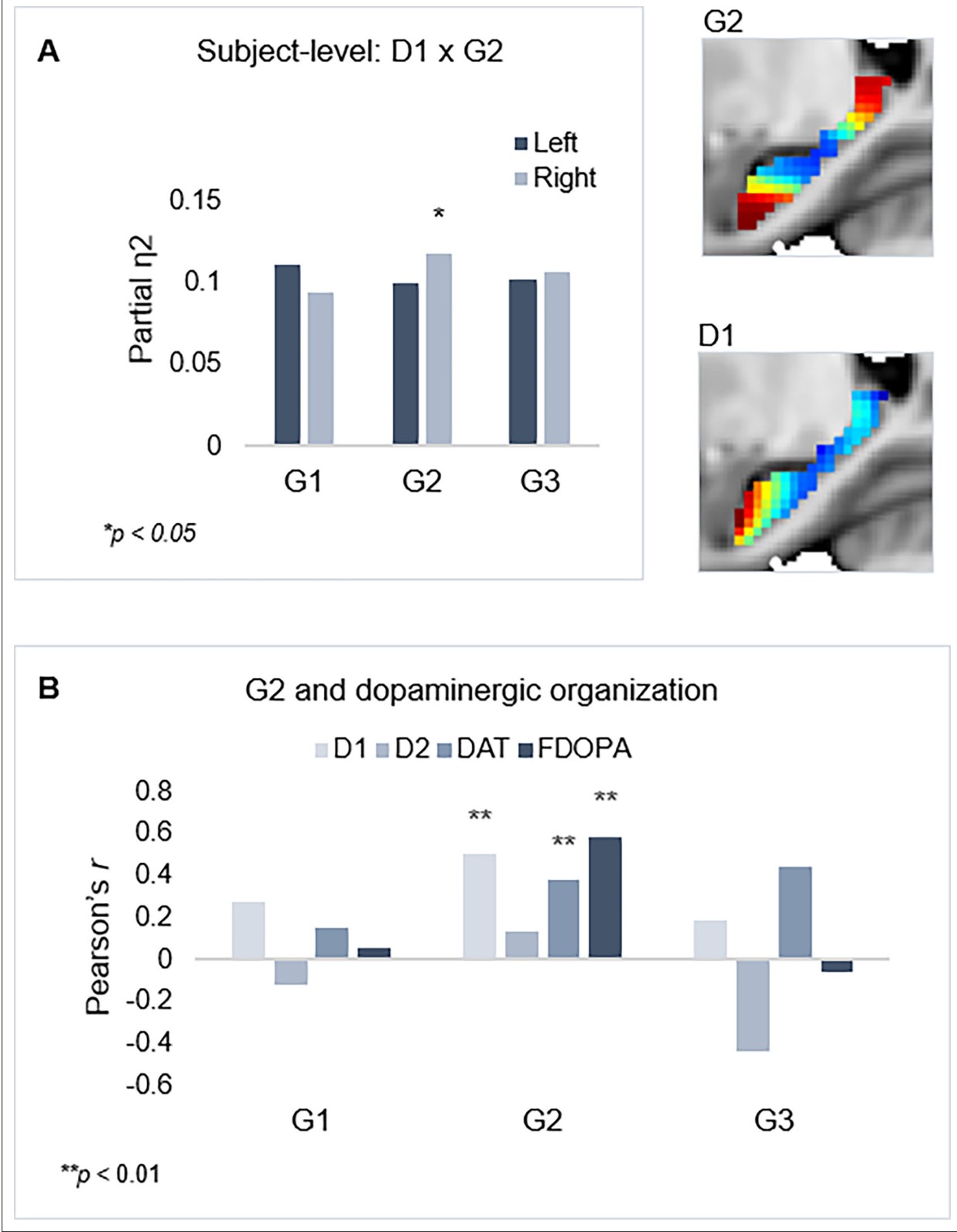

**Figure 3.** Shared functional and molecular organization within the hippocampus. (**A**) Multivariate effects of hippocampal D1DR trend surface modeling (TSM) parameters as predictors of G2 TSM parameters. Images show average organization of G2 and D1DR in the right hemisphere. Note that the arbitrary color scale of G2 has been flipped. (**B**) Correlations between group-level TSM parameters of functional gradients and dopamine (DA) markers.

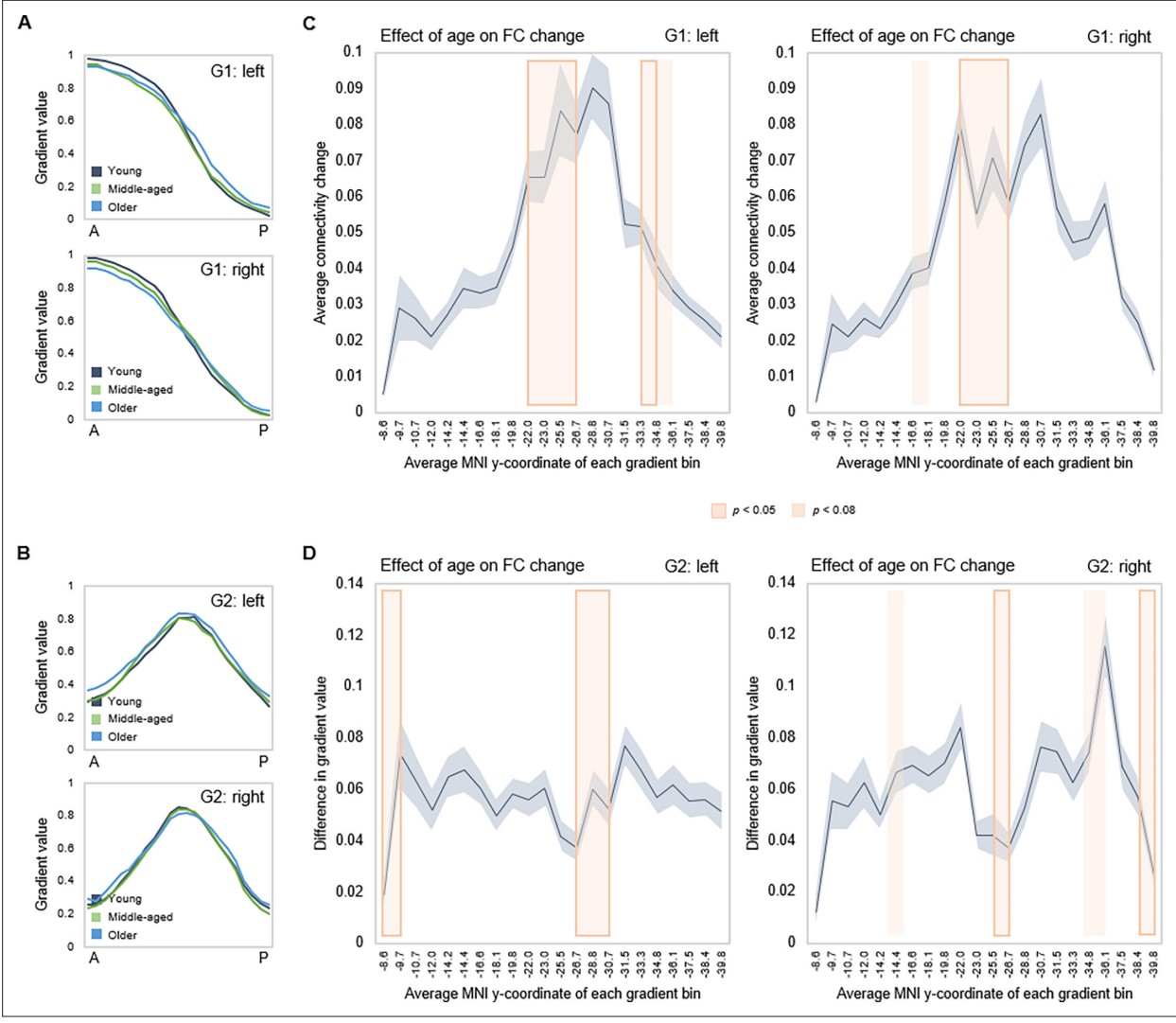

**Figure 4.** Effects of age on hippocampal gradients. (**A, B**) Less specificity in connectivity change across G1 and G2 in older age. Average values of subject-level gradient bins are plotted as a function of their distance from the most anterior hippocampal voxel. Separate lines mark young (20–39 years; gray), middle-aged (40–59 years, green), and older (60–79 years; blue) age-groups. The flatter curves in the older group indicate less distinct change in connectivity patterns across gradients in older age. (**C**) Localization of age effects along G1. Shaded fields indicate the position of significant age effects along G1. (**D**) Localization of age effects along G2. Shaded fields indicate the position of significant age effects along G2.

(20–39 years), middle-aged (40–59 years), and older (60–79 years) adults. Connectivity across G1 and G2 displayed less distinct differentiation at older age, depicted by the flatter curves in the older group (*Figure 4A*). The difference in gradient values between each consecutive pair of gradient bins was computed, and the locations of significant univariate contributions to the effect of age on these values were visualized along gradients (*Figure 4B and C*). For G1, this primarily localized effects to a middle hippocampal region extending posteriorly from just after the uncal apex (MNI y=21).

## Topography of hippocampal gradients predicts episodic memory performance

Next, we tested associations between topography of the three gradients and episodic memory. Using hierarchical multiple regression models in which age, sex, and mean FD were controlled for in a first step (M1), we entered TSM parameters of the three gradients as predictors of episodic memory in a stepwise manner. Models were assessed separately for left and right hemispheres, across the full sample and within age-groups, yielding eight hierarchical models in total. Results are reported with p-values at an uncorrected statistical threshold and p-values after FDR adjustment. Memory

performance was, across the sample, predicted by G2 in the left hippocampus (Adj. $R^2$=0.308, $\Delta R^2$=0.096, F=1.842, p=0.047, $p_{FDR}$ = 0.082) over and above covariates and topography of G1, which did not predict performance (Adj. $R^2$=0.260, $\Delta R^2$=0.029, F=0.695, p=0.713, $p_{FDR}$ = 0.771), and neither did G3 (Adj. $R^2$=0.276, $\Delta R^2$=0.027, F=0.502, p=0.910, $p_{FDR}$ = 0.920) (*Figure 5A*). Observing that the association between G2 and memory did not remain significant after FDR adjustment, we performed the same analysis in our replication dataset, which also included episodic memory testing. Consistent with the observation in our main dataset, G2 significantly predicted memory performance (Adj. $R^2$=0.368, $\Delta R^2$=0.081, F=1.992, p=0.028) over and above covariates and topography of G1. Here, the analysis also showed that G1 topography predicted performance across the sample (Adj. $R^2$=0.325, $\Delta R^2$=0.112, F=3.431, p<0.001).

In our main dataset, memory performance was within young adults, predicted by left-hemisphere G1 (Adj. $R^2$=0.182, $\Delta R^2$=0.357, F=2.672, p=0.015, $p_{FDR}$ = 0.030), whereas neither G2 (Adj. $R^2$=0.204, $\Delta R^2$=0.191, F=1.098, p=0.396, $p_{FDR}$ = 0.492) nor G3 (Adj. $R^2$=0.384, $\Delta R^2$=0.236, F=1.755, p=0.132, $p_{FDR}$ = 0.189) improved the prediction (*Figure 5B*). No gradient predicted memory within middle-aged or older age-groups (F=0.432–1.865, p=0.928–0.113; *Appendix 1—table 1*).

## Youth-like gradient topography supports memory in older age

To investigate the functional role of G1 in old age, the principal and most age-sensitive gradient, we tested whether memory in older adults would be facilitated by youth-like gradient topography using two independent datasets. In DyNAMiC, we applied data-driven latent class analysis (LCA) to TSM parameters (residualized to account for age, sex, and mean FD) of left-hemisphere G1, which predicted episodic memory performance in younger adults. LCA yielded a two-class solution identifying two subgroups (n=19 vs. n=30) of older adults (60–79 years), which by definition differed in left-hemisphere G1 characteristics ($F_{(9,37)}$ = 13.778, p<0.001, $p_{FDR}$ = 0.003, partial $\eta^2$=0.770). A difference was also evident in the right hemisphere ($F_{(9,37)}$ = 3.790, p=0.002, $p_{FDR}$ = 0.005, partial $\eta^2$=0.480).

Individuals in the smaller subgroup were determined as exhibiting an aged gradient profile, whereas older adults in the larger subgroup as exhibiting a youth-like gradient profile. The classification based on G1 parameters extended across all three gradients in both hemispheres (*Figure 6A*), such that the smaller subgroup displayed marked differences from younger adults across all gradients (left G1: $F_{(9,63)}$ = 15.549, p<0.001, $p_{FDR}$ = 0.003, partial $\eta^2$=0.690; right G1: $F_{(9,63)}$ = 5.322, p<0.001, $p_{FDR}$ = 0.003, partial $\eta^2$=0.432; left G2: $F_{(12,60)}$ = 3.991, p<0.001, $p_{FDR}$ = 0.003, partial $\eta^2$=0.444; right G2: $F_{(12,60)}$ = 2.192, p=0.023, $p_{FDR}$ = 0.045, partial $\eta^2$=0.305; left G3: $F_{(12,60)}$ = 2.832, p=0.004, $p_{FDR}$ = 0.01, partial $\eta^2$=0.362; right G3: $F_{(12,60)}$ = 1.844, p=0.061, $p_{FDR}$ = 0.098, partial $\eta^2$=0.269), while the larger subgroup differed less from young adults in terms of G1 (left G1: $F_{(9,74)}$ = 4.416, p<0.001, $p_{FDR}$ = 0.003, partial $\eta^2$=0.349; right G1: $F_{(9,74)}$ = 3.086, p=0.003, $p_{FDR}$ = 0.008, partial $\eta^2$=0.273), and displayed second- and third-order gradients comparable to those in younger age (left G2: $F_{(12,71)}$ = 1.616, p=0.107, $p_{FDR}$ = 0.167, partial $\eta^2$=0.215; right G2: $F_{(12,71)}$ = 1.442, p=0.168, $p_{FDR}$ = 0.235, partial $\eta^2$=0.196; left G3: $F_{(12,71)}$ = 1.122, p=0.357, $p_{FDR}$ = 0.457, partial $\eta^2$=0.159; right G3: $F_{(12,71)}$ = 1.596, p=0.112, $p_{FDR}$ = 0.172, partial $\eta^2$=0.212). See *Appendix 1—figure 7* for classification based on right-hemisphere G1.

Plotting connectivity change along G1 and G2 in the aged and youth-like subgroups revealed that the diminished topographic specificity observed across gradients in older individuals (*Figure 4A*) was driven by older adults with an aged gradient profile (*Figure 6B*). Both older subgroups displayed altered gradient organization across cortex (*Figure 6C and D*). The distribution of cortical networks in G1 space indicated a unimodal-transmodal organization in youth-like older adults, not evident in the aged older group (*Figure 6D*). The two groups did not differ in terms of age (aged: 70.8±6.0; youth-like: 68.4±4.7; t=1.548, p=0.128, $p_{FDR}$ = 0.189), sex (aged: 9 men/10 women; youth-like: 16 men/14 women; $X^2$=0.166, p=0.684, $p_{FDR}$ = 0.746), nor hippocampal gray matter (left hemisphere: aged: 4271.2 ± 480.9 ml; youth-like: 4246.8 ± 269.1 ml; t=0.223, p=0.824, $p_{FDR}$ = 0.850; right hemisphere: aged: 3866.2 ± 446.3 ml; youth-like: 3979.9 ± 398.1 ml; t=0.929, p=0.357, $p_{FDR}$ = 0.457). Subgroups showed comparable levels of average hippocampal D1DR availability (left: youth-like 0.257±0.06; aged 0.242±0.06; t=0.796, p=0.430, $p_{FDR}$ = 0.525; right: youth-like 0.242±0.06; aged 0.251±0.06; t=0.296, p=0.768, $p_{FDR}$ = 0.817) but displayed a pattern of differences in D1DR TSM parameters in comparison to young adults supporting youth-like and aged profiles (youth-like

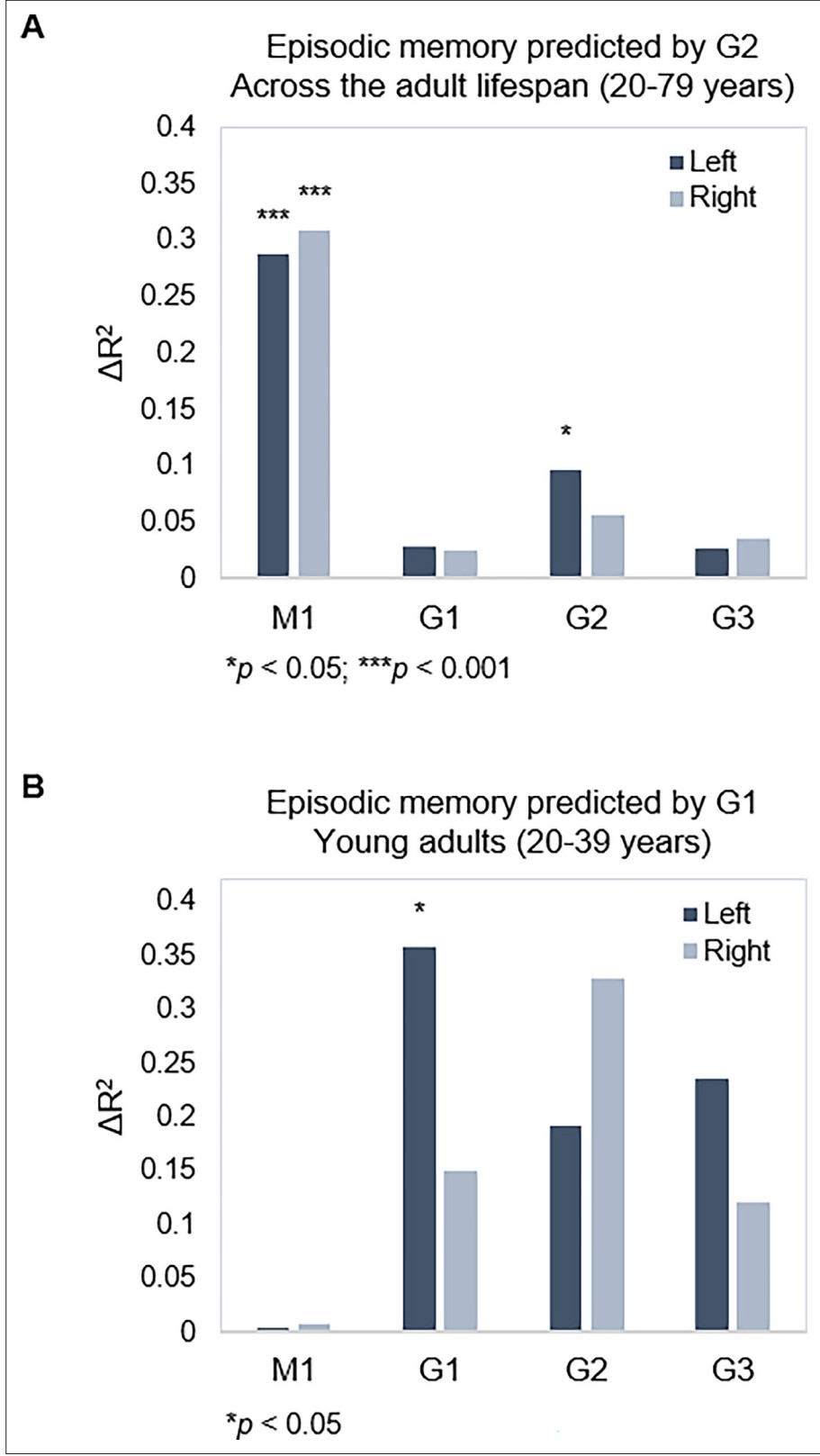

**Figure 5.** Gradient topography and episodic memory. (**A**) Individual differences in topographic characteristics of left-hemisphere G2 significantly predicted episodic memory performance across the sample, over and above the first- and second-step models (M1: age, sex, in-scanner motion; G1 parameters). (**B**) Topographic characteristics of G1 in the left hemisphere significantly predicted episodic memory performance in young adults, over and above M1 (age, sex, and in-scanner motion).

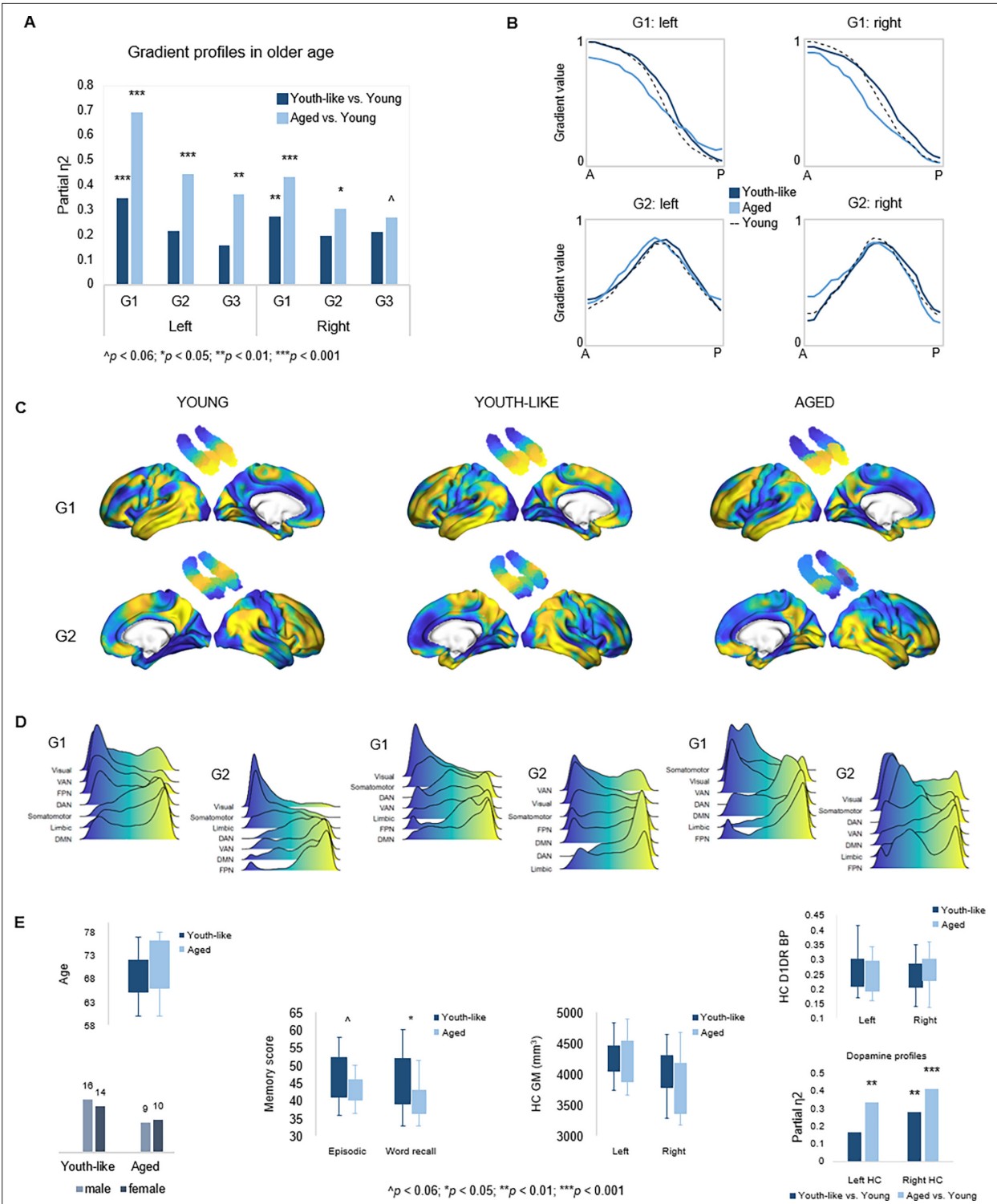

**Figure 6.** Gradient profiles in older age. (**A**) Two older subgroups were identified based on left-hemisphere G1. The first group (n=19) displayed gradient characteristics significantly different from those in young adults, whereas the second group (n=30) displayed gradient characteristics more similar to those in young adults. Bars represent comparisons of gradient trend surface modeling (TSM) parameters between older subgroups and younger adults. (**B**) Average gradient values across participants within subgroups, plotted against the distance (in mm) from the most anterior hippocampal voxel. The flatter curves in older adults with an aged gradient profile suggest less distinct change in connectivity across gradients. (**C**) Group-level G1 and G2 for young, youth-like, and aged groups. (**D**) Position of cortical networks in gradient space. (**E**) Older subgroups were comparable in terms of age, sex, hippocampal gray matter volume, and average levels of hippocampal D1DR availability, while older adults with a youth-like gradient profile exhibited a more youth-like profile also in D1DR distribution and superior episodic memory.

subgroup vs. young: left: $F_{(9,74)}$ = 1.645, p=0.118, $p_{FDR}$ = 0.176, partial $\eta^2$=0.167; aged subgroup vs. young: left: $F_{(9,62)}$ = 3.478, p=0.002, $p_{FDR}$ = 0.005, partial $\eta^2$=0.335; *Figure 6E*).

In line with our hypothesis, we observed superior memory in older adults exhibiting a youth-like gradient profile (*Figure 6E*): at trend level for the composite episodic measure (aged: 43.2±3.7; youth-like: 46.5±6.6; t=1.958, p=0.056, $p_{FDR}$ = 0.092), driven by a significant group difference on its word recall subtest (aged: 40.9±4.5; youth-like: 43.4±6.8; t=2.600, p=0.012, $p_{FDR}$ = 0.025). Word recall performance was furthermore predicted by left-hemisphere G1 parameters (over and above age, sex, and mean FD) in the youth-like older adults (Adj. $R^2$=0.464, $\Delta R^2$=0.543, F=3.043, p=0.028, $p_{FDR}$ = 0.054), while no association was observed in the aged older subgroup (Adj. $R^2$=0.063, $\Delta R^2$=0.533, F=1.004, p=0.518, $p_{FDR}$ = 0.599). Crucially, the ability of left G1 topography to inform classification of older adults into mnemonically distinct subgroups was replicated in the Betula sample (*Appendix 1—figure 7C*).

## Discussion

In this study, we present a comprehensive multidimensional characterization of functional hippocampal-neocortical integration across the adult human lifespan and methodically map topography of connectivity gradients onto behavioral and molecular phenotypes. Our findings reveal the presence of two distinct connectivity gradients distributed along the hippocampus longitudinal axis which contribute to episodic memory function across the lifespan. These observations underscore the importance of disentangling multiple dimensions of hippocampal functional organization in advancing our understanding of cortico-hippocampal systems for memory-related behavior. Moreover, we demonstrate in two independent samples that maintaining a youth-like gradient profile in older age – characterized by preserved distinctiveness of connectivity change along gradients – supports mnemonic functioning, and that increased homogeneity of gradient topography may precede gray matter atrophy.

Despite converging evidence in young adults indicating multiple overlapping modes of functional connectivity across the hippocampus (*Genon et al., 2021*; *Vos de Wael et al., 2018*; *Przeździk et al., 2019*; *Katsumi et al., 2023*; *Tian et al., 2020*), significant questions have remained regarding their spatial distribution across cortex and their role in behavior. Here, connectopic mapping (*Haak et al., 2018b*) identified a principal anteroposterior gradient (G1); a second-order gradient of mid-to-anterior/posterior long-axis variation (G2); and a third-order gradient conveying variation across the hippocampus transverse axis (G3). Although we restrict our discussion of G3 given its low proportion of explained variance, we note that it mirrors patterns previously observed in both structure and function (*Plachti et al., 2019*; *Katsumi et al., 2023*; *Thorp et al., 2022*). Consistently, cortical patterns of G3 matched cortical connectivity profiles previously reported for hippocampal subfields (*Vos de Wael et al., 2018*; *de Flores et al., 2017*). Our results confirm that functional data favors detection of anteroposterior hippocampal organization in contrast to that determined by its cytoarchitecture (*Genon et al., 2021*), but indicate that higher-order connectivity modes may indeed carry coarse-scale information about subfield-determined organization. Greater anatomical specificity, with more precise characterization of connectivity in relation to subfield boundaries while minimizing effects of interindividual differences in hippocampal shape and folding, might be achieved by adopting techniques implementing a geodesic coordinate system to represent effects within the hippocampus (*DeKraker et al., 2022*; *Paquola et al., 2020*).

The general organization of neocortical connectivity within the hippocampus showed stability across the lifespan (*Figure 1*), although clear effects of age were evident in the fine-scale topography of connectivity modes (*Figure 4*). Older age was associated with less distinct transitions in connectivity along G1 and G2, an effect that was exacerbated in a subgroup of older adults exhibiting an aged gradient profile – separated from older adults exhibiting youth-like gradient topography (*Figure 6A and B*). This finding constitutes an important addition to evidence highlighting increased homogeneity in hippocampal function in aging (*Harrison et al., 2019*; *Stark et al., 2021*). Current theories view this loss of specificity as a consequence of its functional isolation from neocortical areas, possibly linked to tau-driven degeneration of the perforant pathway (*Harrison et al., 2019*; *Hyman et al., 1984*). Age-related deterioration of this entorhinal-hippocampal pathway has in turn been linked to impaired mnemonic functioning (*Yassa et al., 2010*). Importantly, older individuals exhibiting dedifferentiated gradient topography, but comparable hippocampal volumes, displayed less efficient episodic memory compared to older adults maintaining youth-like gradient topography (*Figure 6*; *Appendix 1—figure*

*7*). This underscores the potential of gradient-based techniques to capture behaviorally relevant alterations in hippocampal function at a stage preceding structural decline.

Testing the correspondence across cortex of G1 and G2 to canonical gradients of cortical function provided support for G1 reflecting differentiation along a representation-modulation dimension (e.g. task-negative/task-positive [*Fox et al., 2005*]; (*Figure 1C–E*)), separating frontoparietal areas of attention and control from sensorimotor and DMN areas. Consistent with age-related loss of segregation between task-positive and task-negative poles (*Grady et al., 2016*), also evident in numerous diseases involving hippocampal dysfunction (e.g. depression [*Mulders et al., 2015*]), schizophrenia (*Whitfield-Gabrieli et al., 2009*, and AD *Weiler et al., 2017*), we observed altered cortical organization of G1 in both older subgroups. A main difference between subgroups, however, included a unimodal-transmodal organization instead emerging in youth-like older adults, whereas not in the aged subgroup (*Figure 6D*). A meaningful role of this potential reorganization was indicated by an association between G1 topography and memory in the youth-like older adults only. In terms of G2, we provide support of its characterization as a local representation of the principal unimodal-transmodal cortical gradient (*Figure 1C–E*), widely demonstrated across functional, structural, and molecular modalities (*Huntenburg et al., 2018*; *Margulies et al., 2016*; *Hansen et al., 2021*). Taken together, our observations support a framework of cortico-hippocampal integration in which the DMN is positioned in different neural contexts: in one case at the opposite end from frontoparietal networks of attention and control, and in the other, opposite to sensorimotor and visual networks – indicating that an account of the hippocampus functional connectivity with the DMN is dependent on multiple neurofunctional hierarchies. The overlap of G1 and G2 may potentially reflect the superimposition of gradients and hubs of long-axis anatomical connections indicated in both the human and animal hippocampus (*Strange et al., 2014*; *Dalton et al., 2022*). Moreover, the observation that macro-scale relationships between distinct cortical systems are mapped out by G1 and G2 in this manner may reflect the hippocampus primordial position in the laminar development of the cerebral cortex (*Goulas et al., 2019*), supporting the idea that hippocampal function stands, from a phylogenetic perspective, to inform general principles of brain organization (*Genon et al., 2021*).

We discovered that G2, specifically, manifested organizational principles shared among function, behavior, and neuromodulation. Meta-analytical decoding reproduced a unimodal-associative axis across G2 (*Figure 3B*), and analyses in relation to the distribution of D1DRs – which vary across cortex along a unimodal-transmodal axis (*Froudist-Walsh et al., 2021*; *Pedersen et al., 2023a*) – demonstrated topographic correspondence both at the level of individual differences and across the group. It should, however, be acknowledged that PET imaging in the hippocampus is associated with resolution-related limitations, although previous research indicates high test-retest reliability of [¹¹C] SCH23390 PET to quantify D1DR availability in this region (*Kaller et al., 2017*). As such, mapping the distribution of hippocampal D1DRs at a fine spatial scale remains challenging, and replication of our results in terms of overlap with G2 is needed in independent samples. Here, we evaluated the observed spatial overlap between G2 topography and D1DRs across multiple TSM orders, showing correspondence between modalities from simple to more complex parameterizations of their spatial properties. Topographic correspondence was additionally observed between G2 and other DA markers from independent datasets (*Figure 3B*), suggesting that G2 may constitute a mode reflecting a dopaminergic phenotype, which contributes to the currently limited understanding of its biological underpinnings.

Results linked both G1 and G2 to episodic memory, suggesting complimentary contributions of these two overlapping long-axis modes. Considered together, analyses in the main and replication datasets indicated a role of G2 topography in memory across the adult lifespan, independent of age. A similar association with G1 was only evident across the entire sample in the replication dataset, whereas results in the main sample seemed to emphasize a role of youth-like G1 topography in memory performance. In line with previous research, memory was successfully predicted by G1 topography in young adults (*Przeździk et al., 2019*) and similarly predicted by G1 in older adults exhibiting a youth-like functional profile.

It is noteworthy that meta-analytical decoding of G2 primarily linked the unimodal connectivity patterns of anterior and posterior subregions to terms of episodic memory, encoding, and navigation (*Figure 2C*). G2's role in memory might as such be considered in light of hippocampal integration with the visual system, by which it contributes to complex perceptual processes supporting memory

(*Turk-Browne, 2019*). In humans, there is evidence of direct hippocampal connections to early visual areas, with recent tractography-based work demonstrating connectivity as primarily localized to the posterior hippocampus and to a smaller region in the anterior hippocampus (*Dalton et al., 2022*).

The verbal nature of our memory tasks likely contributed to the left-lateralization of effects, yet, predominant left-hemisphere vulnerability to aging and age-related pathology should not be ruled out as a meaningful contributor to these effects (*Minkova et al., 2017*). Average hippocampal D1DR availability did not differ between older subgroups, but a tendency toward youth-like and aged functional profiles being mirrored in D1DR topography was observed (*Figure 6E*). However, longitudinal data is ultimately required to inform the underlying mechanisms of individual differences in hippocampal gradient topography in older age (*Nyberg et al., 2012*; *Cabeza et al., 2018*). Future studies should, furthermore, assess gradients' modulation by behavioral conditions and extend these methods to clinical samples characterized by hippocampal dysfunction.

This study establishes behavioral relevance of two overlapping long-axis modes of hippocampal-neocortical functional connectivity, shedding light on their age-related dedifferentiation, and its impact on cognition. In sum, this study introduces a multidimensional framework for understanding hippocampal-neocortical integration and its interplay with memory and neuromodulation throughout the adult human lifespan.

## Materials and methods

This study included data from the DyNAMiC study, for which the design and procedures have been described in detail elsewhere (*Nordin et al., 2022*; *Johansson et al., 2023*). Here, we include the materials and methods relevant to the current study. DyNAMiC was approved by the Regional Ethical Board and the local Radiation Safety Committee of Umeå, Sweden. All participants provided written informed consent prior to testing.

### Participants

The DyNAMiC sample included 180 participants (20–79 years; mean age = 49.8 ± 17.4; 90 men/90 women equally distributed within each decade). Individuals were randomly selected from the population register of Umeå, Sweden, and recruited via postal mail. Exclusion criteria implemented during the recruitment procedure included brain pathology, impaired cognitive functioning (Mini-Mental State Examination<26), medical conditions and treatment that could affect brain functioning and cognition (e.g. dementia, diabetes, and psychiatric diagnosis), and brain imaging contraindications (e.g. metal implants). All participants were native Swedish speakers. A total of 16 participants were excluded from connectopic mapping due to excessive in-scanner motion, leaving resting-state fMRI data for 164 participants (20–78 years; mean age = 48.7 ± 17.3). As a replication dataset, we used an independent sample of 224 cognitively healthy and native Swedish-speaking adults (122 men/102 women; 29–85 years mean age = 65.0 ± 13.0) from the population-based Betula project, for which the design and recruitment procedures have been reported in detail elsewhere (*Nilsson et al., 2004*).

### Episodic memory

Episodic memory was measured using three tasks testing word recall, number-word recall, and object-location recall, respectively (*Nordin et al., 2022*). In the word recall task, participants were presented with 16 Swedish concrete nouns that appeared successively on a computer screen. Each word was presented for 6 s during encoding with an interstimulus interval (ISI) of 1 s. Following encoding, participants reported as many words as they could recall by typing them using the keyboard. Two trials were completed, yielding a maximum score of 32. In the number-word task, participants encoded pairs of 2-digit numbers and concrete plural nouns (e.g. 46 dogs). During encoding, eight number-word pairs were presented, each displayed for 6 s, with an ISI of 1 s. Following encoding, nouns were presented again, in a re-arranged order, and participants had to report the 2-digit number associated with each presented noun (e.g. How many dogs?). This task included two trials with a total maximum score of 16. The third task was an object-location memory task. Here, participants were presented with a 6×6 square grid in which 12 objects were, one by one, shown at distinct locations. Each object-position pairing was displayed for 8 s, with an ISI of 1 s. Following encoding, all objects were simultaneously

shown next to the grid for the participant to move them (in any order) to their correct position in the grid. If unable to recall the correct position of an object, participants had to guess and place the object in the grid to the best of their ability. Two trials of this task were completed, making the total maximum score 24.

A composite score of performances across the three tasks was calculated and used as the measure of episodic memory. For each of the three tasks, scores were summarized across the total number of trials. The three resulting sum scores were z-standardized and averaged to form one composite score of episodic memory performance (T score: mean = 50; SD = 10). Missing values were replaced by the average of the available observed scores.

## Image acquisition

Brain imaging was conducted at Umeå University Hospital, Sweden. Structural and functional MRI data were acquired with a 3T Discovery MR 750 scanner (General Electric, WI, USA), using a 32-channel head coil. PET data were acquired with a Discovery PET/CT 690 scanner (General Electric, WI, USA).

### Structural MRI

Anatomical T1-weighted images were acquired with a 3D fast-spoiled gradient-echo sequence, collected as 176 slices with a thickness of 1 mm. Repetition time (TR) was 8.2 ms, echo-time (TE)=3.2 ms, flip angle = 12°, and field of view (FOV)=250 × 250 mm$^2$.

### Functional MRI

Functional MR data were collected during resting state, with participants instructed to keep their eyes open and focus on a fixation cross during scanning. Images were acquired using a T2*-weighted single-shot echo-planar imaging sequence, with a total of 350 volumes collected over 12 min. The functional time series was sampled with 37 transaxial slices, slice thickness = 3.4 mm, and 0.5 mm spacing, TR = 2000 ms, TE = 30 ms, flip angle = 80°, and FOV = 250 × 250 mm$^2$. Ten dummy scans were collected at the start of the sequence.

### PET imaging

PET was conducted in 3D mode with a Discovery PET/CT 690 (General Electric, WI, USA) to assess whole-brain DA D1 receptor availability using the radioligand [$^{11}$C]SCH23390. Scanning was done during a resting condition, with participants instructed to lay still and remain awake with their eyes open. To minimize head movement, a thermoplastic mask (Posicast; CIVCO Medical Solutions; IA, USA) was individually fitted for each participant and attached to the bed surface during scanning. Following a low-dose CT scan (10 mA, 120 kV, and 0.8 s rotation time) for attenuation correction, an intravenous bolus injection with target radioactivity of 350 MBq [$^{11}$C]SCH23390 was administered. The PET scan was a 60 min dynamic scan, with 6×10 s, 6×20 s, 6×40 s, 9×60 s, 22×120 s frames. The average radioactivity dose administered to participants was 337±27 MBq (range 205–391 MBq). Due to participant dropout and technical issues, complete PET data was available for 177 DyNAMiC participants.

## Image preprocessing
### Hippocampal segmentation and volumetric assessment

Individual anatomical T1-weighted images were submitted to automated segmentation in FreeSurfer version 6 (*Fischl et al., 2002*). A mean image of participants' normalized T1-weighted images was also segmented in FreeSurfer, and yielded hippocampal and cortical segmentations used as masks for connectopic mapping. Regional gray matter volume was estimated from subject-specific hippocampal segmentations and were corrected for total intracranial volume (ICV; the sum of volumes for gray matter, white matter, and cerebrospinal fluid). Adjusted volumes were equal to the raw volume – b(ICV – mean ICV), where b is the regression slope of volume on ICV (*Buckner et al., 2004*). Automated segmentation of the hippocampus into subiculum, CA1–3, and DG/CA4 subfields was conducted in FreeSurfer using the group-average T1-weighted image, for sample-specific masks to overlay onto G3 (*Appendix 1—figure 2*).

## fMRI data

Resting-state fMRI data were preprocessed using Statistical Parametric Mapping (SPM12: Wellcome Trust Centre for Neuroimaging, http://www.fil.ion.ucl.ac.uk/spm/) implemented in an in-house software, DataZ. Functional images were slice-timing corrected, co-registered to the anatomical T1-images, and motion corrected, and underwent distortion correction using subject-specific B0-field maps. The functional data were subsequently co-registered to the anatomical T1-images again, temporally demeaned, and linear and quadratic effects were removed. Next, a 36-parameter nuisance regression model was applied (*Ciric et al., 2017*), including mean cerebrospinal, white matter, and whole-brain signal in addition to six motion parameters, including parameters' squares, derivatives, and squared derivatives. To further control for in-scanner motion, the model also included a set of spike regressors, defined as binary vectors of motion-contaminated volumes exceeding a volume-to-volume root-mean-squared (RMS) displacement of 0.25 mm. A temporal high-pass filter (with a threshold of 0.009 Hz) was applied simultaneously as nuisance regression in order to not re-introduce nuisance signals. Finally, images were normalized to MNI space by Diffeomorphic Anatomical Registration using Exponentiated Lie algebra (DARTEL [*Ashburner, 2007*]) and smoothed with a 6 mm full-width-at-half-maximum (FWHM) Gaussian kernel. Four individuals were excluded from the template-generation step due to non-pathological anatomical irregularities. In total, 16 participants were excluded due to displaying excessive in-scanner motion, as defined by displaying (i) more than 20 volumes with >0.25 relative RMS difference in motion and (ii) greater than 0.2 average RMS across the run. On average, the relative RMS difference in motion across the sample was 0.090 (±0.063), and the mean FD was 0.164 (±0.104).

## DA D1 receptor availability

Preprocessing of PET data was performed in SPM12 (Wellcome Trust Centre for Neuroimaging, http://www.fil.ion.ucl.ac.uk/spm/). BP relative to non-displaceable binding in a reference region (BP$_{ND}$; *Innis et al., 2007*) was used as an estimate of receptor availability (i.e. D1DR) in the hippocampus, for each participant defined using the FreeSurfer segmentation of their anatomical images. Cerebellum was used as reference region. PET images were corrected for head movement by using frame-to-frame image co-registration and co-registered with T1-weighted MRI images with re-slicing to T1 voxel size. The simplified reference-tissue model was used to model regional time-activity course (TAC) data. Regional TAC data were adjusted for partial volume effects by using the symmetric geometric transfer matrix method implemented in FreeSurfer and an estimated point spread function of 2.5 mm FWHM. We additionally used data from a publicly available database of group-averaged volumetric maps of molecular target distributions (https://github.com/netneurolab/hansen_receptors; *Hansen, 2022*). Specifically, we downloaded previously published maps of DAT (*Dukart et al., 2018*) and FDOPA (https://www.nitrc.org/projects/spmtemplates) to investigate the spatial correspondence between functional gradients and dopaminergic target distributions.

## Mapping gradients of functional connectivity

Connectopic mapping (*Haak et al., 2018b*) was run through the ConGrads toolbox (*Haak et al., 2018b*) implemented in FSL (*Smith et al., 2004*). Mapping was conducted on both subject level and group level, for the left and right hippocampus separately, and involved two main steps. First, for every hippocampal voxel, connectivity fingerprints were computed as the Pearson correlation between the voxel-wise time series and a singular-value decomposition representation of all cortical voxels. In a second step, nonlinear manifold learning (Laplacian eigenmaps) was applied to a matrix expressing the degree of similarity between the voxel-wise fingerprints. This yields eigenvectors, so called connectopic maps, representing modes of functional connectivity (i.e. functional gradients). Each connectopic map is then projected onto cortex, for which each vertex is color coded according to the voxel in the hippocampus it correlates the most with. Since connectopic mapping at group level involves applying Laplacian eigenmaps to a group-average similarity matrix, group-level mapping across the sample was conducted using the hippocampal and cortical masks derived from the FreeSurfer segmentation of a sample-mean structural image. Mapping was specified to compute 20 gradients, and a subsequent scree plot over explained variance indicated meaningful contributions of the first three connectopic maps, together explaining 67% of the variance across hemispheres (*Appendix 1—figure 1*).

## Stability of functional gradients across levels of spatial smoothing

It has recently been suggested that the reliability of connectopic mapping to detect meaningful gradients of resting-state functional connectivity may be limited due to spatial smoothing implemented during preprocessing of fMRI data (*Watson and Andrews, 2023*). To determine the stability of our hippocampal gradients, we conducted a series of control analyses across varying levels of smoothing, and in contrast to connectopic maps derived through connectopic mapping on random data. These analyses are presented in Appendix 1 and confirmed high stability of resting-state gradients and their ability to capture interindividual differences, whereas random data failed to produce meaningful gradients (*Appendix 1—figure 4*).

## Alignment of connectopic maps across participants

To ensure optimal alignment of identified connectopic maps across participants, we employed Procrustes alignment, based on voxel-wise correlations, to order subject-level connectopic maps according to their correspondence with a set of reference maps (i.e. gradients computed at group level across the full sample). Moreover, whereas the sign of connectopic maps is arbitrary, differences therein have an impact on the spatial model parameters describing the topographic characteristics of gradients, derived through TSM in a later step. As such, the sign of subject-level connectopic maps showing negative correlations with the corresponding group-level reference map were inversed.

## Trend surface modeling

Using spatial statistics, the topography of a connectopic map can be represented by a small number of spatial model parameters. This parameterization enables analyses of interindividual differences and is achieved through TSM, implemented in a third step of the ConGrads analysis pipeline (*Haak et al., 2018b*). In this step, the spatial pattern of each subject-level connectopic map is approximated by estimating a spatial statistical model. Model estimation involves fitting a set of polynomial basis functions along canonical axes of the connectopic map. In MNI space, this entails estimation along x, y, and z axes of the hippocampus. Thus, fitting a polynomial of degree 1 yields three TSM parameters (x, y, z), with any increase in model order corresponding to an increase in number of parameters (e.g. six parameters for the second model order: $x$, $y$, $z$, $x^2$, $y^2$, $z^2$; nine parameters for the third model order, etc.). TSMs are fitted with an increasing polynomial degree using Bayesian linear regression, which provides likelihood estimates that can be used for subsequent model selection. Here, model selection was based on three information sources: (i) the Bayesian information criterion (BIC) across subjects for models estimated at orders 1–10; (ii) the % explained variance in connectopic maps by each model; and (iii) visual inspection of group-level gradients reconstructed from TSM parameters at different model orders. The purpose of using multiple information sources, as opposed to simply BIC, was to find a trade-off between high-quality reconstructions of gradients by TSMs, while keeping the number of model parameters sufficiently low for multivariate statistical analyses. A model order of 3 (=9 TSM parameters) was selected for G1, whereas a model order of 4 (=12 TSM parameters) was selected for G2 and G3 (*Appendix 1—figure 5*). Each gradient's set of TSM parameters were then used as either dependent or independent variables in multivariate GLMs investigating links between gradient topography and variables such as age, episodic memory performance, and D1DR distribution.

## Transitions in connectivity as a function of the hippocampal longitudinal axis

To visualize the orthogonal patterns of change in connectivity conveyed by each gradient and to aid in the interpretation of age effects, we divided each subject-level connectopic map into 23 bins of ~2 mm along the hippocampus anterior-posterior axis and estimated the average gradient value (ranging from 0 to 1) for each bin. Plotting the values of each bin against their distance in mm from the most anterior voxel in the hippocampus as such demonstrates the pattern of change in connectivity along the anterior-posterior axis (*Prze̹dzik et al., 2019*).

## Cortical projections and correlations with gradients of cortical function

In ConGrads (*Haak et al., 2018b*), cortical mapping is computed independently for each gradient based on the regression equation: pmap = $\mathbf{B^T}.(\mathbf{X^{T-1}}.\mathbf{A^T})^\top$, where A corresponds to data inside (i.e. hippocampus) and B to data outside (i.e. cortical) the ROI, and X-1 is the pseudo-inverse of the

corresponding eigenvector (stacked with a row of ones). For each hippocampal gradient, the volumetric MNI-registered group-level cortical projection map was resampled to the Human Connectome Project 32k_LR midthickness surface mesh, using the volume-to-surface algorithm with enclosing mapping available in Connectome Workbench v.1.5.0. Spearman correlations were computed between the cortical projection of each hippocampal gradient and the three gradients previously established as the main axes of cortical functional organization (*Margulies et al., 2016*). To reduce the degree of freedom, each surface projection was resampled by 1000-cortical parcels (*Schaefer et al., 2018*). Statistical significance of correlations was assessed by spin-test permutation (*Alexander-Bloch et al., 2018*), randomly rotating a spherical projection of the cortical maps 1000 times, with two-tailed statistical significance determined at a 95% confidence level.

## Mapping behavioral transitions along gradients using Neurosynth

Transitions in behavioral domains were mapped onto G1 and G2 using meta-analytical decoding in Neurosynth (*Yarkoni et al., 2011*). We assessed two sets of behavioral terms (*Figure 2*), the first was a selection of terms commonly linked to anteroposterior hippocampal functional specialization (*Plachti et al., 2019*; *Grady, 2020*), and the second a selection of terms based on a previous report demonstrating behavioral transitions along a unimodal-transmodal cortical axis (*Margulies et al., 2016*). For correspondence with meta-analytical maps, we created region of interest masks by projecting the cortical surface of each gradient to the 2 mm volumetric MNI152 standard space. These volumetric images were then divided into five 25 percentile bins and binarized. The resulting images were used as input to the Neurosynth decoder, yielding an r statistic associated with each behavioral term per section of each gradient.

## Acknowledgements

This work was supported by the Swedish Research Council (grant number 2016-01936 to AS), Riksbankens Jubileumsfond (grant number P20-0515 to AS), Knut and Alice Wallenberg Foundation (Wallenberg Fellow grant to AS), and StratNeuro grant at Karolinska Institutet (AS). The Betula Study is supported by a Scholar grant to LN from the Knut and Alice Wallenberg Foundation. FreeSurfer calculations were enabled by resources provided by the Swedish National Infrastructure for Computing (SNIC) at HPC2N, partially funded by the Swedish Research Council through grant agreement no. 2018-05973.

## Additional information

### Funding

| Funder | Grant reference number | Author |
| --- | --- | --- |
| Vetenskapsrådet | 2016-01936 | Alireza Salami |
| Riksbankens Jubileumsfond | P20-0515 | Alireza Salami |
| Knut och Alice Wallenbergs Stiftelse | Fellow grant | Alireza Salami |
| Karolinska Institutet | StratNeuro | Alireza Salami |
| Knut och Alice Wallenbergs Stiftelse | Scholar grant | Lars Nyberg |

The funders had no role in study design, data collection and interpretation, or the decision to submit the work for publication.

### Author contributions

Kristin Nordin, Conceptualization, Formal analysis, Investigation, Visualization, Methodology, Writing – original draft, Writing – review and editing; Robin Pedersen, Jarkko Johansson, Micael Andersson, Formal analysis, Investigation, Visualization, Writing – review and editing; Farshad Falahati, Formal

analysis, Investigation, Writing – review and editing; Filip Grill, Anna Rieckmann, Conceptualization, Investigation, Writing – review and editing; Saana M Korkki, Lars Bäckman, Andrew Zalesky, Lars Nyberg, Investigation, Writing – review and editing; Alireza Salami, Conceptualization, Formal analysis, Funding acquisition, Investigation, Writing – original draft, Writing – review and editing

**Author ORCIDs**
Kristin Nordin ⓘ https://orcid.org/0000-0003-4157-1638
Jarkko Johansson ⓘ https://orcid.org/0000-0002-4501-4735
Lars Nyberg ⓘ https://orcid.org/0000-0002-3367-1746
Alireza Salami ⓘ https://orcid.org/0000-0002-4675-8437

**Ethics**
Human subjects: This study was approved by the Regional Ethical board and the local Radiation Safety Committee in Umeå, Sweden (approval number 2017/248-31). All participants provided written informed consent prior to testing.

Reviewer #2 (Public review): https://doi.org/10.7554/eLife.97658.3.sa1
Reviewer #3 (Public review): https://doi.org/10.7554/eLife.97658.3.sa2
Author response https://doi.org/10.7554/eLife.97658.3.sa3

---

## Additional files

**Supplementary files**
MDAR checklist

**Data availability**
Data from the DyNAMiC study cannot be made publicly available. Swedish and European data protection legislation and the conditions of the ethical approval (which does not include data sharing) restricts sharing of data from the DyNAMiC study. These restrictions apply to original data, deidentified data, and to source data of graphs in this manuscript. Access to the data may be granted for research without commercial purposes upon formal request from and evaluation by the Principal Investigator of the DyNAMiC study, Dr Alireza Salami (alireza.salami@ki.se), Aging Research Center, Karolinska Institutet, Sweden, and would require a completed data transfer agreement. The Matlab, R, and FSL codes used for analyses included in this study are openly available at https://github.com/kristinnordin/hcgradients (copy archived at *Nordin, 2025*). Computation of gradients was done using the freely available toolbox ConGrads: https://github.com/koenhaak/congrads (*Haak et al., 2018a*).

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

## Appendix 1

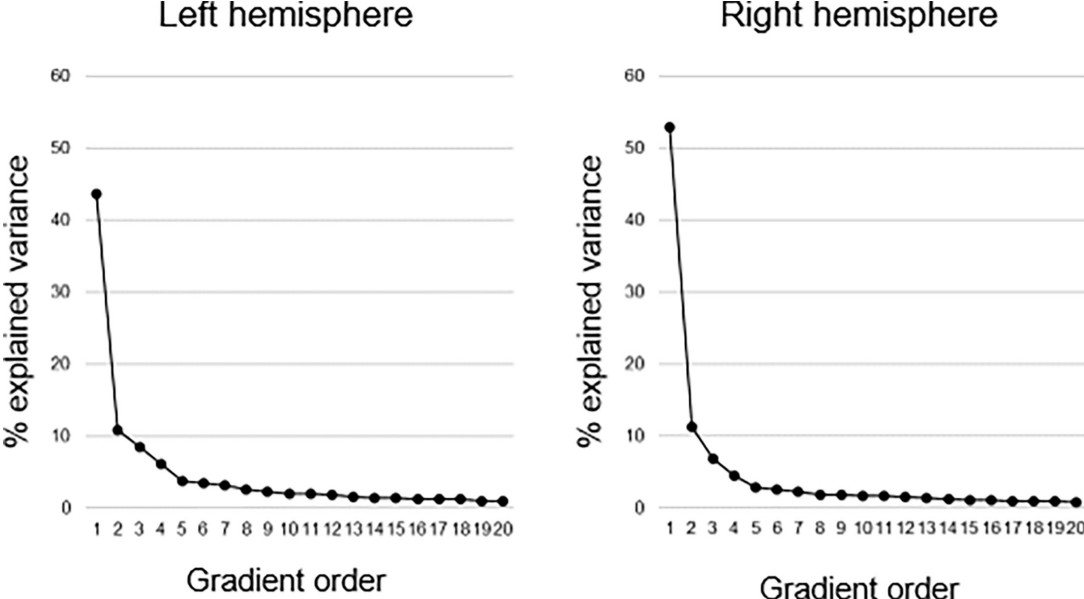

**Appendix 1—figure 1.** Explained variance of gradients. Connectopic mapping was used to compute 20 hippocampal gradients in each hemisphere on group level across the sample. The first-order gradient explained 44% and 53% of the variance in left and right hemispheres, respectively. The second-order gradient explained 11% in both hemispheres, and the third-order gradient 8% and 7% in left and right hemispheres, respectively.

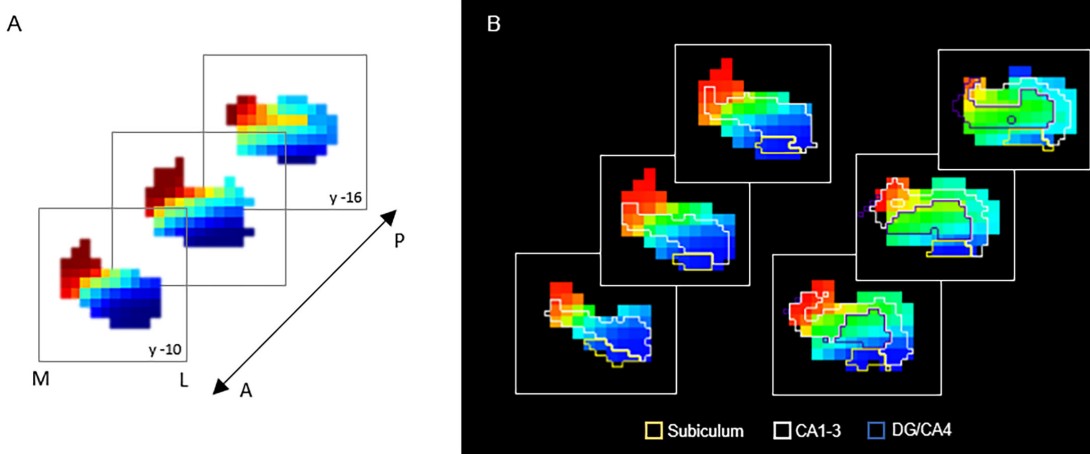

**Appendix 1—figure 2.** G3 in relation to hippocampal subfields. (**A**) Coronal view of the third-order connectopic map, G3, within the anterior hippocampus, where medial-lateral variation was most evident. (**B**) Hippocampal subfields are displayed as contours: subiculum (yellow), CA1–3 (white), DG/CA4 (blue).

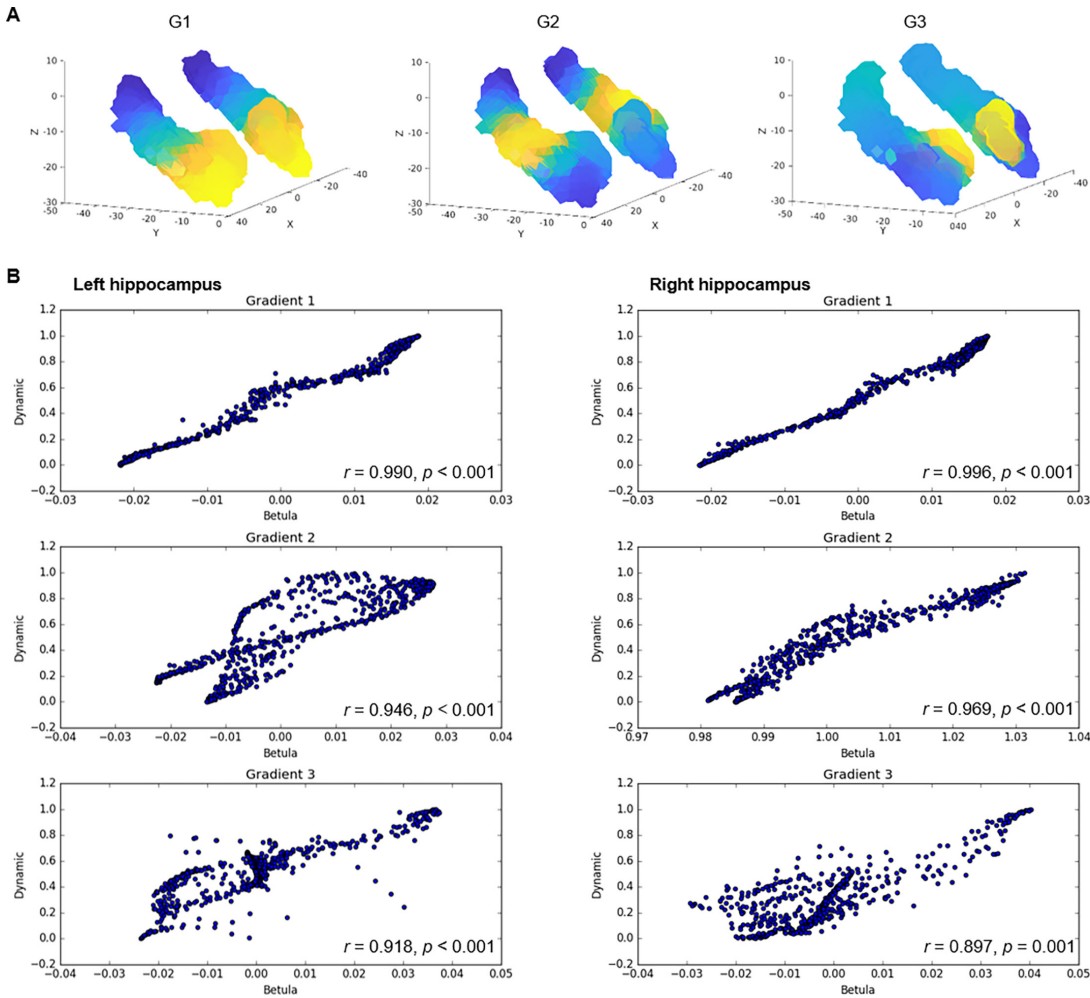

**Appendix 1—figure 3.** Connectopic mapping in an independent dataset. (**A**) Gradients of hippocampal cortical connectivity were replicated in an independent sample of 224 adults (122 men/102 women; 29–85 years, mean age = 65.0 ± 13.0) from the Betula project (**Nilsson et al., 2004**; **Nyberg et al., 2020**), such that a principal anterior-posterior gradient (G1) was followed by a second-order middle-to-anterior/posterior gradient (G2) and a third-order inferior-lateral to medial gradient (G3). (**B**) Voxel-wise correlations between gradients in the Dynamic and Betula datasets.

## Stability of gradients across levels of spatial smoothing

Spatial smoothing is commonly implemented during preprocessing of fMRI data. Benefits of smoothing include enhancing the signal-to-noise ratio and reducing effects of anatomical variability between subjects (**Molloy et al., 2014**; **Wu et al., 2011**). On the other hand, smoothing is associated with greater spatial autocorrelation. In a recent study, **Watson and Andrews, 2023**, suggested that spatial autocorrelation induced by smoothing during preprocessing is sufficient for detecting functional gradients through connectopic mapping, even in artificially generated random fMRI time series (**Watson and Andrews, 2023**). To determine the stability of our hippocampal gradients, in light of smoothing as a potential confound, we ran connectopic mapping on both true and random data at varying levels of spatial smoothing (i.e. no smoothing, 0.5 mm and 6.0 mm).

Random fMRI data were generated according to the methods specified by **Watson and Andrews, 2023**. Briefly, we synthesized Gaussian white noise matched in mean and variance to the real fMRI data in native space. The random time series therefore approximate the signal amplitude and variation of the true data, without a coherent spatial or temporal correlation structure. The random fMRI data were then normalized to MNI space with varying levels of smoothing. Normalization for both true and random data was performed using DARTEL (**Ashburner, 2007**) implemented in SPM12. This method projects the location of voxels to a template, effectively preserving the tissue

count, but is known to introduce aliasing artifacts if the original data is at a similar or lower resolution than the deformation fields. The effects of aliasing artifacts are typically minimized by smoothing the data during the normalization procedure. For non-smoothed data, missing voxels along the X and Y dimensions were replaced by interpolating neighboring voxels along the Z dimension. This effectively removes aliasing artifacts by resampling a single voxel along the projected dimension, avoiding spurious autocorrelations otherwise produced by smoothing.

Our results support the detection of expected hippocampal gradients in true data independent of smoothing degree (*Appendix 1—figure 4A and B*). Age-related variance was captured in both left and right G1 at all levels of smoothing, whereas additionally visible across G2 and G3 from a minimum level of 0.5 mm smoothing (*Appendix 1—figure 4C*). In contrast, cmaps derived from random data showed poor consistency with expected gradients (*Appendix 1—figure 4B*) and could not capture individual differences (*Appendix 1—figure 4C*). In sum, the hippocampal gradients in our work are highly stable in their spatial layout, and in their ability to capture interindividual differences, across levels of spatial smoothing. Observations in random data indicate that spatial smoothing is not sufficient to produce meaningful gradients of functional variance in the hippocampus.

**Appendix 1—table 1.** Gradient topography as a predictor of episodic memory performance.

| | Full sample | | Young (20–40 years) | | Middle-aged (40–60 years) | | Older (60–79 years) | |
|---|---|---|---|---|---|---|---|---|
| | $\Delta R^2$ | F | $\Delta R^2$ | F | $\Delta R^2$ | F | $\Delta R^2$ | F |
| Model 1* | 0.287 | 20.92*** | 0.003 | 0.058 | 0.131 | 2.567^ | 0.163 | 2.915* |
| *Left HC* | | | | | | | | |
| G1 | 0.029 | 0.695 | 0.357 | 2.672* | 0.076 | 0.450 | 0.156 | 0.916 |
| G2 | 0.096 | 1.842* | 0.191 | 1.098 | 0.264 | 1.246 | 0.215 | 0.921 |
| G3 | 0.027 | 0.502 | 0.236 | 1.755 | 0.261 | 1.462 | 0.256 | 1.219 |
| *Right HC* | | | | | | | | |
| G1 | 0.026 | 0.621 | 0.150 | 0.810 | 0.128 | 0.804 | 0.169 | 0.987 |
| G2 | 0.056 | 1.013 | 0.328 | 1.542 | 0.221 | 1.066 | 0.151 | 0.566 |
| G3 | 0.036 | 0.633 | 0.120 | 0.432 | 0.288 | 1.865 | 0.276 | 1.070 |

*M1=age, sex, mean frame-wise displacement; ^p<0.07; *p<0.05; **p<0.01; ***p<0.001.

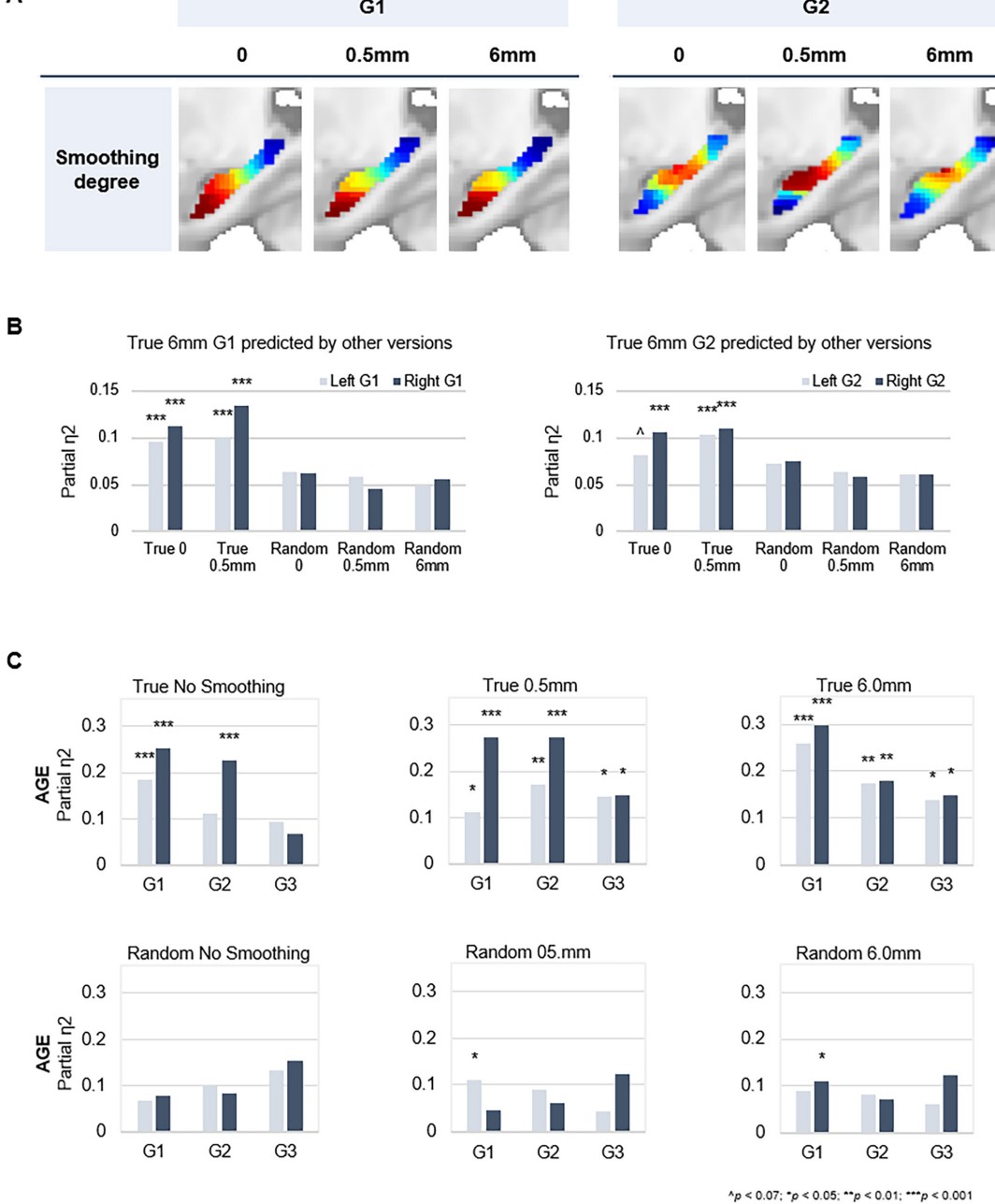

**Appendix 1—figure 4.** Connectopic mapping on true and random data at varying levels of spatial smoothing. (**A**) G1 (left panel) and G2 (right panel) connectopic maps derived from resting-state functional magnetic resonance imaging (fMRI) data after different degrees of spatial smoothing. (**B**) Bars represent effect sizes from multivariate GLMs with trend surface modeling (TSM) parameters of the G1 (left) and G2 (right) connectopic maps derived from 6.0 mm smoothed data (presented in the main text) as dependent variables, predicted by TSM parameters of connectopic maps based on (i) true data not smoothed; (ii) true 0.5 mm smoothed data; (iii) random data not smoothed; (iv) random 0.5 mm smoothed data; (v) random 6.0 mm smoothed data. (**C**) Effects of age on the topographic characteristics of gradients (i.e. TSM parameters) for true and random versions across levels of spatial smoothing.

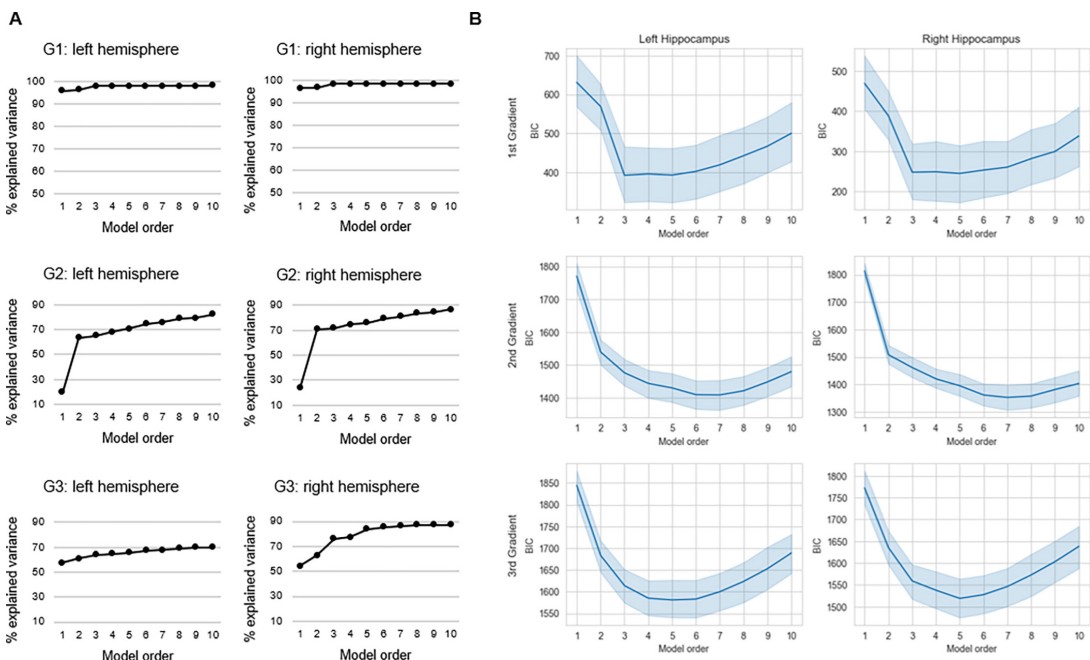

**Appendix 1—figure 5.** Selection of trend surface modeling (TSM) order. (**A**) The % explained variance in group-level connectopic maps by TSMs. (**B**) Average values of the Bayesian information criterion (BIC) across participants are plotted against TSM orders. Shaded areas represent the 95% confidence interval.

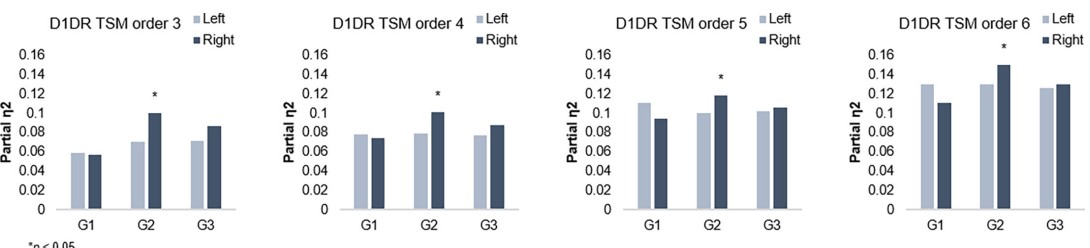

**Appendix 1—figure 6.** G2 and dopamine D1 receptor distribution. Individual differences in D1DR topography predict G2 topography in the right hemisphere across trend surface modeling (TSM) orders.

## Classification of older adults based on right-hemisphere G1 parameters

Classification of older adults based on right-hemisphere G1 TSM parameters yielded a two-class solution, by definition, these two groups differed from each other in terms of right-hemisphere G1 characteristics ($F_{(9,37)}$ = 10.886, p<0.001, partial $\eta^2$=0.726), with a difference between groups also evident in the left hemisphere ($F_{(9,37)}$ = 2.248, p=0.040, partial $\eta^2$=0.353). The two subgroups primarily showed a youth-like (vs. young: $F_{(9,73)}$ = 1.428, p=0.192, partial $\eta^2$=0.150) vs. aged (vs. young: $F_{(9,70)}$ = 8.574, p<0.001, partial $\eta^2$=0.524) gradient profile in terms of right-hemisphere G1 parameters (*Appendix 1—figure 6A*), as such, the classification did not extend across the other gradients like the one based on left-hemisphere G1 parameters. The two subgroups did not significantly differ in terms of age (aged: 70.0±5.8; youth-like: 68.5±5.0; t=1.030, p=0.308), sex (aged: 13 men/10 women; youth-like: 13 men/13 women; $X^2$=0.208, p=0.648) nor hippocampal gray matter volume (left hemisphere: aged: 4345.4±448.4; youth-like: 4173.1±269.0; t=1.614, p=0.114; right hemisphere: aged: 3981.2±428.2; youth-like: 3874.4±426.9; t=0.872, p=0.388). Subgroups identified based on right-hemisphere G1 topography did not significantly differ in episodic memory performance (the composite episodic measure used across the sample: aged: 43.9±5.01; youth-like: 45.6±6.0; t=1.093, p=0.280; word recall subtest: aged: 42.0±5.2; youth-like: 44.9±7.8; t=1.545, p=0.129).

## Replication of older classes in an independent sample

The ability of left G1 topography to inform classification of older adults into mnemonically distinct subgroups was replicated in the Betula sample (class 1: n=60; class 2: n=99; $F_{(9,149)}$ = 35.993, p<0.001, partial $\eta^2$=0.685), *Appendix 1—figure 7C*. A difference in gradient topography between classes was also evident for G1 in the right hemisphere ($F_{(9,149)}$ = 2.134, p=0.030, partial $\eta^2$=0.114) but was more limited across subsequent gradients in both hemispheres. The smaller class was, in relation to young adults, exhibiting an aged G1 ($F_{(9,63)}$ = 4.166, p<0.001, partial $\eta^2$=0.373) in contrast to the bigger class displaying a more youth-like G1 ($F_{(9,102)}$ = 1.345, p=0.223, partial $\eta^2$=0.106). Importantly, we observed a trend-level difference in word recall performance between subgroups (t(157) = 1.585, p=0.058, one-tailed), such that youth-like older adults displayed superior memory performance.

**Classification of older adults based on right-hemisphere G1**

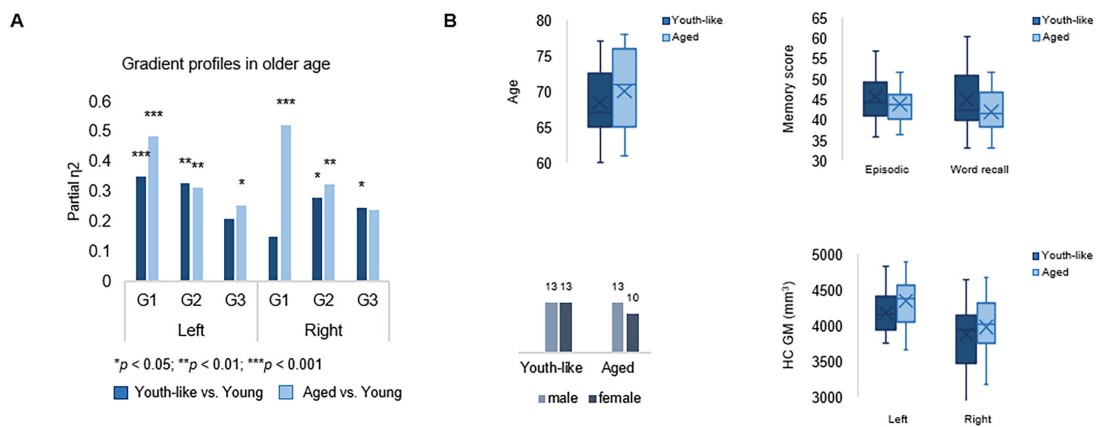

**Older adults with youth-like and aged gradient profiles identified in an independent sample**

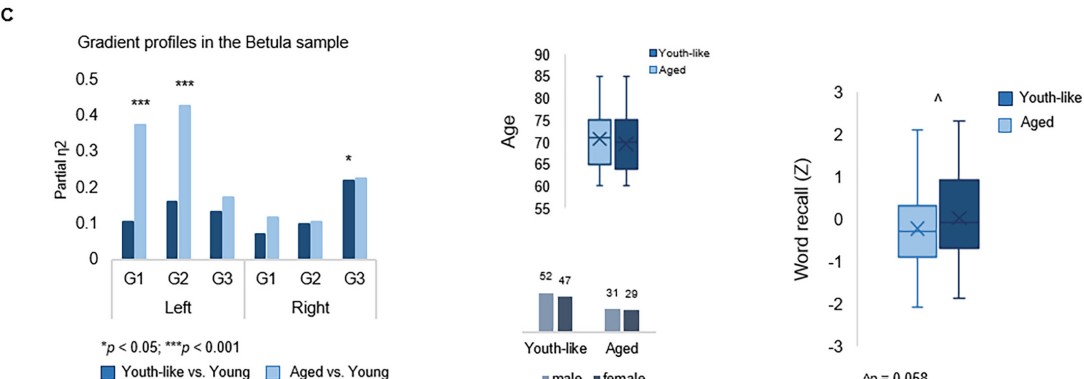

**Appendix 1—figure 7.** Classification of older adults based on gradient topography. (**A**) Classification of older adults in DopamiNe, Age, connectoMe, and Cognition (DyNAMiC) based on right-hemisphere G1 parameters. The first group (n=23) displayed right G1 characteristics significantly different from those in young adults, whereas the second group (n=26) displayed right G1 characteristics more similar to those in young adults. (**B**) Older subgroups were comparable in terms of age, sex, hippocampal gray matter (GM) volume, and memory performance. (**C**) Classification of older adults in the independent sample, Betula, based on left-hemisphere G1. A subgroup with a youth-like G1 displayed higher episodic memory performance.

