## [Editor Report · eLife Assessment]

This **fundamental** work demonstrates the importance of considering overlapping modes of functional organization (i.e. gradients) in the hippocampus, showing associations with aging, dopaminergic receptor distribution and episodic memory. The evidence supporting the conclusions is **convincing**, although not all analyses were performed in a replication sample. The work will be of broad interest to basic and clinical neuroscientists.

---

## [Referee Report · Reviewer #2 (Public review)]

Summary:

This paper derives the first three functional gradients in the left and right hippocampus across two datasets. These gradient maps are then compared to dopamine receptor maps obtained with PET, associated with age, and linked to memory. Results reveal links between dopamine maps and gradient 2, age with gradients 1 and 2, and memory performance.

Strengths:

This paper investigates how hippocampal gradients relate to aging, memory, and dopamine receptors, which are interesting and important questions. A strength of the paper is that some of the findings were replicated in a separate sample.

Assessment after revision:

The authors addressed concerns about unclear multiple comparison correction in the revision. The replication sample was primarily used to replicate the topographic organization of functional hippocampal-neocortical connectivity within the hippocampus across the adult lifespan, which was the central goal of this paper. Not all other analyses replicated, which the authors nicely clarified in the revised manuscript. Overall, this work is a thorough and valuable contribution to the literature.

---

## [Referee Report · Reviewer #3 (Public review)]

Summary:

In this study, the authors analyzed the complex functional organization of the hippocampus using two separate adult lifespan datasets. They investigated how individual variations in the detailed connectivity patterns within the hippocampus relate to behavioral and molecular traits. The findings confirm three overlapping hippocampal gradients and reveal that each is linked to established functional patterns in the cortex, the arrangement of dopamine receptors within the hippocampus, and differences in memory abilities among individuals. By employing multivariate data analysis techniques, they identified older adults who display a hippocampal gradient pattern resembling that of younger individuals and exhibit better memory performance compared to their age-matched peers. This underscores the behavioral importance of maintaining a specific functional organization within the hippocampus as people age.

Strengths:

The evidence supporting the conclusions is compelling, based on a unique dataset, a rich set of carefully unpacked results, and a rigorous data analysis that is clearly explained and motivated. Possible confounds are carefully considered and ruled out.

Assessment after revision:

The authors improved the transparency of the statistical analyses by stating explicitly what tests and corrections were performed and clearly justifying the elected statistical approaches. They now also acknowledge and discuss the potential limitations of the presented PET analyses. Overall this is a rigorous and important contribution to the literature that will likely be of broad interest to basic and clinical neuroscience.

---

## [Author Response]

The following is the authors’ response to the original reviews.

**Public Reviews:**

**Reviewer #1 (Public Review):**
The authors studied how hippocampal connectivity gradients across the lifespan, and how these relate to memory function and neurotransmitter distributions. They observed older age with less distinct transitions and observed an association between gradient de-differentiation and cognitive decline.This is overall an innovative and interesting study to assess gradient alterations across the lifespan and its associations to cognition.The paper is well-written, and the methods appear sound and thoughtful. There are several strengths, including the inclusion of two independent cohorts, the use of gradient mapping and alignment techniques, and an overall sound statistical and analysis framework. There are several areas for potential improvements in the paper, and these are listed below:

We thank the Reviewer for their positive assessment and summary of our work. We address each of the Reviewer’s comments below, and outline the revisions we have made to the manuscript based on the Reviewer’s suggestions.

(1) The reported D1 associations appear a bit post-hoc in the current work and I was unclear why the authors specifically focussed on dopamine here, as other transmitter systems are similar present at the level of the hippocampus and implicated in aging.

Other neurotransmitter systems may indeed be relevant in the context of hippocampal function in aging. In this study, however, we included a specific research question about the DA D1 receptor (D1DR) based on previous research (1) emphasizing the role of DA neuromodulation in maintaining functional network segregation in aging to support cognition (Pedersen et al., 2023), (2) reporting heterogeneous distribution of DA markers across the hippocampus, supporting efficient modulation of distinct behaviors (Dubovyk & ManahanVaughan, 2019; Edelmann & Lessmann, 2018; Gasbarri et al., 1994; Kempadoo et al., 2016), and (3) demonstrating the spatial distribution of D1DRs as varying across neocortex along a unimodal-transmodal gradient (Pedersen et al., 2024). To which degree this variation might be reflected in cortico-hippocampal connectivity, however, remained to be investigated. As such, one of the study’s specific aims was to evaluate the spatial distribution of D1DRs as a molecular correlate of the hippocampus’ functional organization. Importantly, we were interested in mapping associations between individual differences in the organization of connectivity and D1DRs. This was uniquely enabled by utilizing the DyNAMiC sample, as it includes structural and functional MRI data in combination with D1DR PET in the same individuals across the adult lifespan (n=180). However, after observing significant spatial correspondence between functional organization and D1DR expressed by the second hippocampal gradient (G2), we did indeed perform complimentary analyses with group-averaged data of additional dopamine markers (D2DR from a subsample of our participants, as well as DAT and FDOPA from open sources) to test the generalizability of the original finding. Taken together, the original analyses based on subject-level data and complimentary group-level analyses provided support for the interpretation of G2 as a dopaminergic mode.

We have updated the manuscript to clarify the focus on the D1 receptor and the contribution of including additional DA markers.

Updated paragraph in the Introduction, pages 5-6:

“Dopamine (DA) is one of the most important modulators of hippocampus-dependent function(47,48), and influences the brain’s functional architecture through enhancing specificity of neuronal signaling(49). Consistently, there is a DA-dependent aspect of maintained functional network segregation in aging which supports cognition(50). Animal models suggest heterogeneous patterns of DA innervation(51,52) and postsynaptic DA receptors(53), across both transverse and longitudinal hippocampal axes, likely allowing for separation between DA modulation of distinct hippocampus-dependent behaviors(47). Moreover, the human hippocampus has been linked to distinct DA circuits on the basis of long-axis variation in functional connectivity with midbrain and striatal regions(54,55). Taken together with recent findings revealing a unimodal-transmodal organization of the most abundantly expressed DA receptor subtype, D1 (D1DR), across cortex(56), we tested the hypothesis that the organization of hippocampal-neocortical connectivity partly reflects the underlying distribution of hippocampal DA receptors, predicting predominant spatial correspondence for any hippocampal gradient conveying a unimodal-transmodal pattern across cortex.”

Updated sections in the Results, page 13-14:

“Our next aim was to investigate to which extent the distribution of hippocampal DA D1 receptors (D1DRs), measured by [^11^C]SCH23390 PET in the DyNAMiC(58) sample, may serve as a molecular correlate of the hippocampus’ functional organization.”

“Complimentary analyses were then conducted to further evaluate G2 as a dopaminergic hippocampal mode by utilizing additional DA markers at group-level.”

Moreover, the authors may be aware that multiple PET tracers are somewhat challenged in the mesiotemporal region. Is this the case for the D1 receptor as well? The hippocampus is a small and complex structure, and PET more of a low res technique so one would want to highlight and discuss the limitations of the correlations with PET maps here and/or evaluate whether the analysis adds necessary findings to the study.

We thank the Reviewer for raising this point. The lower resolution of PET is indeed a relevant aspect to consider when quantifying D1DR availability in the hippocampus, even though previous research indicate high test-retest reliability of [^11^C]SCH23390 PET measurement in this region (Kaller et al., 2017). We have now elaborated on PET limitations in the Discussion of the revised manuscript.

In our study, we made efforts to reduce potential partial volume effects (PVE) by correcting our PET data, and tested spatial associations between our functional gradients and D1DR maps using trend-surface modelling (TSM), rather than through voxel-wise comparisons. This allowed us to evaluate the spatial correspondence between functional connectivity and D1DRs at a level of spatial trends, estimated using TSM models computed at increasing levels of complexity. The results showed consistent spatial overlap between G2 and D1DRs across these models, that is, across spatial trends described at coarser-to-finer scales. Furthermore, this was replicated across several DA markers with PET and SPECT data from independent samples.

Taken together, we agree with the Reviewer that the spatial correspondence observed between G2 and hippocampal D1DRs should be interpreted in the context of resolution-related limitations inherent to PET imaging. However, we strongly believe that our DA analyses offer valuable insight to the molecular underpinnings of hippocampal functional organization.

Updated paragraph in the Discussion, pages 25-26:

“We discovered that G2, specifically, manifested organizational principles shared among function, behavior, and neuromodulation. Meta-analytical decoding reproduced a unimodalassociative axis across G2 (Figure 3B), and analyses in relation to the distribution of D1DRs – which vary across cortex along a unimodal-transmodal axis(76,77) – demonstrated topographic correspondence both at the level of individual differences and across the group. It should, however, be acknowledged that PET imaging in the hippocampus is associated with resolutionrelated limitations, although previous research indicate high test-retest reliability of [^11^C]SCH23390 PET to quantify D1DR availability in this region(78). As such, mapping the distribution of hippocampal D1DRs at a fine spatial scale remains challenging, and replication of our results in terms of overlap with G2 is needed in independent samples. Here, we evaluated the observed spatial overlap between G2 topography and D1DRs across multiple TSM model orders, showing correspondence between modalities from simple to more complex parameterizations of their spatial properties. Topographic correspondence was additionally observed between G2 and other DA markers from independent datasets (Figure 3B), suggesting that G2 may constitute a mode reflecting a dopaminergic phenotype, which contributes to the currently limited understanding of its biological underpinnings.”

From my (perhaps somewhat biased) perspective, it might be valuable to instead or in addition look at measures of hippocampal microstructure and how these relate to the functional aging effects. This could be done, if available, using data from the same subjects (eg based on quantitative MRI contrasts and/or structural MRI) and/or using contextualization findings as implemented in eg hippomaps.readthedocs.io

We thank the Reviewer for this suggestion. We performed additional analyses investigating the spatial overlap between our connectivity gradients and estimates of hippocampal microstructure, computed as the ratio of T1- over T2-weighted (T1w/T2w) images (Glasser & Von Essen, 2011; vos de Wael et al., 2018). Analyses of spatial correspondence then followed the TSM-based method used to test the spatial overlap between functional connectivity gradients and D1DR distribution. Applying TSM to the T1w/T2w image computed for each participant yielded subject-level model parameters describing microstructure topography, which were then entered as predictors of connectivity topography in multivariate GLMs (separate models for each gradient and hemisphere, 6 models in total).

Analyses revealed that microstructure of the right hippocampus significantly predicted gradient topography of right-hemisphere G1 (F = 1.325, *p* = 0.034), while no other links between connectivity gradients and microstructure emerged as significant (F 0.930-1.184, *ps* 0.7060.079).

These results, suggesting an association along the anteroposterior axis, deviate from previous findings linking hippocampal microstructure to G3-like, medial-lateral, connectivity organization (vos de Wael et al., 2018). As we believe that comprehensive analyses of our gradients in relation to microstructure across the lifespan would be best addressed in future work, we have not included these analyses of microstructure in the revised manuscript.

(2) Can the authors clarify why they did not replicate based on cohorts that are more widely used in the community and open access, such as CamCAN and/or HCP-Aging? It might connect their results with other studies if an attempt was made to also show that findings persist in either of these repositories.

We agree with the Reviewer that replication in samples such as CamCAN and/or HCP-Aging would provide valuable opportunities to connect our findings with those of other studies using those datasets. Here, we included the Betula dataset (Nilsson et al., 2004) as our replication sample, as it was immediately available to us, included a large sample of adults in a comparable age, and a word recall episodic memory task closely aligned with the one included in DyNAMiC. Importantly, leveraging the Betula dataset as our replication sample allows us to link our findings to a wide range of previous studies central to the understanding of neurocognitive aging in general, and hippocampal aging in particular (Nyberg, 2017; Nyberg et al., 2020). Betula is a large longitudinal project that has been tracking individuals since 1988, and is part of the National E-infrastructure for Aging Research (NEAR: www.near-aging.se), through which data from several Swedish studies are made available to both national and international researchers. While we acknowledge the value of extending replication efforts to datasets like CamCAN and HCP-Aging, we emphasize the significant contribution of having replicated our connectivity gradients in the Betula dataset.

(3) The authors applied TSM and related these parameters to topographic changes in the gradients. I was wondering whether and how such an approach controls for autocorrelation present in both the PET map and gradients. Could the authors clarify?

The Reviewer raises an important topic in spatial autocorrelation. The TSM approach used to parameterize the topography of the functional gradients and D1DR distribution, and to test the spatial correspondence between modalities, did not include any specific method to control for autocorrelation. Here, we highlight two aspects of our study in relation to this point. First, we demonstrated in the Supplementary information (S. Figure 4) that autocorrelation induced by spatial smoothing likely has limited effects on overall gradient topography and the ability of TSM parameters to capture meaningful inter-individual differences in terms of age. Second, in the case of spatial overlap effects being significantly impacted by autocorrelation, we would expect the association between right-hemisphere G2 and D1DR topography to similarly emerge for G2 in the left hemisphere. The absence of such an association may speak to a limited effect of spatial autocorrelation.

(4) The TSM approach quantifies the gradients in terms of x/y/z direction in a cartesian coordinate system. Wouldn't a shape intrinsic coordinate system in the hippocampus also be interesting, and perhaps even be more efficient to look at here (see eg DeKraker 2022 eLife or Paquola et al 2020 eLife)?

This is a very relevant question and we appreciate the Reviewer’s suggestion. We recognize that there may be several benefits associated with adopting a shape-intrinsic coordinate system when characterizing effects in the hippocampus, given its curved/folded anatomy. Approaches like the ones adopted in DeKraker et al., 2022 and Paquola et al., 2020, utilizes geodesic coordinate frameworks to represent the hippocampus in surface space, enabling mapping of connectivity onto the hippocampal surface while respecting its inherent curvature and topology. We anticipate that quantifying gradients within such a framework would especially benefit identification of connectivity change across the hippocampal surface relative to reference points such as subfield boundaries, while minimizing effects of interindividual differences in hippocampal shape and folding. In our study, hippocampal gradients and their associated cortical patterns were computed in volumetric space, with TSM subsequently used to parameterize the change in connectivity along these gradients. This indeed yields a description of connectivity change within a coordinate system less specific to hippocampal anatomy, but may favor generalizability and integration with previous gradient findings within and beyond the hippocampus (e.g., Przeździk et al., 2019; Tian et al., 2020; Katsumi et al., 2023; Navarro-Schröder et al., 2015), as well as connections with broader neuroimaging frameworks through techniques such as meta-analytical decoding. In our view, the different coordinate frameworks offer complimentary insight to hippocampal organization, and while we have opted to not undertake novel analyses to explore our gradients within a geodesic coordinate system for the purposes of this paper, we recognize the importance of such evaluation of our gradients in future analyses. We have made updates to the Discussion in the revised manuscript on this topic (pages 23-24):

“Greater anatomical specificity, with more precise characterization of connectivity in relation to subfield boundaries while minimizing effects of inter-individual differences in hippocampal shape and folding, might be achieved by adopting techniques implementing a geodesic coordinate system to represent effects within the hippocampus(68,69).”

**Reviewer #2 (Public Review):**
Summary:This paper derives the first three functional gradients in the left and right hippocampus across two datasets. These gradient maps are then compared to dopamine receptor maps obtained with PET, associated with age, and linked to memory. Results reveal links between dopamine maps and gradient 2, age with gradients 1 and 2, and memory performance.Strengths:This paper investigates how hippocampal gradients relate to aging, memory, and dopamine receptors, which are interesting and important questions. A strength of the paper is that some of the findings were replicated in a separate sample.Weaknesses:The paper would benefit from added clarification on the number of models/comparisons for each test. Furthermore, it would be helpful to clarify whether or not multiple comparison correction was performed and - if so - what type or - if not - to provide a justification. The manuscript would furthermore benefit from code sharing and clarifying which results did/did not replicate.

We thank the Reviewer for their positive assessment and suggestions regarding further clarifications. We have addressed the Reviewer’s comments in a point-by-point manner under the “Recommendations for the authors” section.

**Reviewer #3 (Public Review):**
Summary:In this study, the authors analyzed the complex functional organization of the hippocampus using two separate adult lifespan datasets. They investigated how individual variations in the detailed connectivity patterns within the hippocampus relate to behavioral and molecular traits. The findings confirm three overlapping hippocampal gradients and reveal that each is linked to established functional patterns in the cortex, the arrangement of dopamine receptors within the hippocampus, and differences in memory abilities among individuals. By employing multivariate data analysis techniques, they identified older adults who display a hippocampal gradient pattern resembling that of younger individuals and exhibit better memory performance compared to their age-matched peers. This underscores the behavioral importance of maintaining a specific functional organization within the hippocampus as people age.Strengths:The evidence supporting the conclusions is overall compelling, based on a unique dataset, rich set of carefully unpacked results, and an in-depth data analysis. Possible confounds are carefully considered and ruled out.Weaknesses:No major weaknesses. The transparency of the statistical analyses could be improved by explicitly (1) stating what tests and corrections (if any) were performed, and (2) justifying the elected statistical approaches. Further, some of the findings related to the DA markers are borderline statistically significant and therefore perhaps less compelling but they line up nicely with results obtained using experimental animals and I expect the small effect sizes to be largely related to the quality and specificity of the PET data rather than the derived functional connectivity gradients.

We thank the Reviewer for the thoughtful summary and positive assessment of our work. To increase transparency of the statistical analyses, we have in the revised manuscript added information regarding statistical tests and corrections for multiple comparisons. In the Results, p-values were reported at an uncorrected statistical threshold, and we have in the revised manuscript included the corresponding p-values adjusted for multiple comparisons using the Benjamini-Hochberg method to control the false discovery rate (FDR). Finally, in the revised manuscript, we have now elaborated on the potential limitations of our PET analyses and we include the updated paragraph below.

Addition made to the Results section, page 13:

“Individual maps of D1DR binding potential (BP) were also submitted to TSM, yielding a set of spatial model parameters describing the topographic characteristics of hippocampal D1DR distribution for each participant. D1DR parameters were subsequently used as predictors of gradient parameters in one multivariate GLM per gradient (in total 6 GLMs, controlled for age, sex, and mean FD). Results are reported with p-values at an uncorrected statistical threshold and p-values after adjustment for multiple comparisons using the Benjamini-Hochberg method to control the false discovery rate (FDR).”

Addition made to the Results section, page 15:

“Effects of age on gradient topography were assessed using multivariate GLMs including age as the predictor and gradient TSM parameters as dependent variables (controlling for sex and mean frame-wise displacement; FD). One model was fitted per gradient and hemisphere, each model including all TSM parameters belonging to a gradient (in total, 6 GLMs).”

Addition made to the Results section, page 17:

“Models were assessed separately for left and right hemispheres, across the full sample and within age groups, yielding eight hierarchical models in total. Results are reported with p-values at an uncorrected statistical threshold and p-values after FDR adjustment.”

Updated paragraph in the Discussion, pages 25-26:

“We discovered that G2, specifically, manifested organizational principles shared among function, behavior, and neuromodulation. Meta-analytical decoding reproduced a unimodalassociative axis across G2 (Figure 3B), and analyses in relation to the distribution of D1DRs – which vary across cortex along a unimodal-transmodal axis(76,77) – demonstrated topographic correspondence both at the level of individual differences and across the group. It should, however, be acknowledged that PET imaging in the hippocampus is associated with resolutionrelated limitations, although previous research indicate high test-retest reliability of [^11^C]SCH23390 PET to quantify D1DR availability in this region(78). As such, mapping the distribution of hippocampal D1DRs at a fine spatial scale remains challenging, and replication of our results in terms of overlap with G2 is needed in independent samples. Here, we evaluated the observed spatial overlap between G2 topography and D1DRs across multiple TSM model orders, showing correspondence between modalities from simple to more complex parameterizations of their spatial properties. Topographic correspondence was additionally observed between G2 and other DA markers from independent datasets (Figure 3B), suggesting that G2 may constitute a mode reflecting a dopaminergic phenotype, which contributes to the currently limited understanding of its biological underpinnings.”

**Recommendations for the authors:**

**Reviewer #1 (Recommendations For The Authors):**
Please see the comments in the public review.

We thank the Reviewer for their comments and recommendations, and have addressed them in the “Public review” section.

**Reviewer #2 (Recommendations For The Authors):**
(1) All statistical analyses are based on linear regressions using trend surface modeling (TSM) parameters that parameterize gradients at the subject level. These models resulted in 9 parameters for gradient 1 and 12 parameters each for gradients 2 and 3. The text states that 'Effects of age on gradient topography was assessed using multivariate GLMs including age as the predictor and gradient TSM parameters as dependent variables (controlling for sex and mean frame-wise displacement; FD)'. Please clarify whether these GLMs were fitted separately for each TSM parameter (i.e., 9+12+12=33 models for both left and right = 66 total models) or on the overall model?

We appreciate the Reviewer’s request for clarification on this matter. These GLMs were fitted on the overall TSM model, that is, through one GLM per gradient (3) and hemisphere (2), each one including all TSM parameters belonging to a gradient (in total, 6 GLMs).

In the revised manuscript, we have added more details to the Results section, page 15: “Effects of age on gradient topography were assessed using multivariate GLMs including age as the predictor and gradient TSM parameters as dependent variables (controlling for sex and mean frame-wise displacement; FD). One model was fitted per gradient and hemisphere, each model including all TSM parameters belonging to a gradient (in total, 6 GLMs).”

(2) Similarly, for memory it appears that multiple models were performed (left and right, young, middle-aged, old, whole groups). Please clarify whether and how multiple comparison correction was performed in this case.

In the revised manuscript, we have now specified the number of analyses conducted in relation to memory performance. We have also clarified that p-values were reported at an uncorrected statistical threshold, and we have in the revised manuscript included the corresponding p-values adjusted for multiple comparisons using the Benjamini-Hochberg method to control the FDR.

Updated section in the Results, page 17:

“Models were assessed separately for left and right hemispheres, across the full sample and within age groups, yielding eight hierarchical models in total. Results are reported with p-values at an uncorrected statistical threshold and p-values after FDR adjustment.”

(3) Although I applaud the authors for their replication efforts, the results do not appear to replicate well. For example, memory was linked to gradient 2 in the whole group but to gradient 1 in the young group. Furthermore, dopamine was linked to gradient 2 in the right but not the left hemisphere. Although the overall group-level gradients were very stable between the two datasets, it is not clear whether the age findings replicated and the memory subgroup findings only replicated at trend level for memory and only partially replicated at the TSM parameter level.

We thank the Reviewer for highlighting the inclusion of a replication dataset as a strength of our study, and we appreciate the recommendation to clarify to which extent results replicated. We provide a response to the Reviewer’s points below, and specify the revisions made to the manuscript in relation to this topic.

The main aim of our study was to characterize the topographic organization of functional hippocampal-neocortical connectivity within the hippocampus across the adult lifespan, as previous studies have limited their focus to younger adults. Given the lack of previous studies for comparison, together with our identification of a novel secondary long-axis connectivity gradient (G2) taking precedence over the previously established medial-lateral G3, we included the Betula sample (Nilsson et al., 2004) for the purpose of replication. There was a high level of consistency between our main dataset and our replication dataset, with gradients 1-3 in left and right hemispheres identified in both samples.

Further use of the replication dataset, beyond the identification of the connectivity gradients, was originally not planned. As such, not all subsequent analyses in the main dataset were conducted in the replication dataset. However, we found it critical to evaluate the observation that older individuals who maintained a youth-like gradient topography also exhibited higher levels of memory performance in an independent sample. This was possible given that the replication dataset included a comparable number of participants in similar ages and a word recall episodic memory task corresponding well to the one used in DyNAMiC. Overall, we conclude that these analyses replicated well across samples. Firstly, topography of lefthemisphere G1 informed the classification of older adults into youth-like and aged subgroups in both samples. Furthermore, in both samples, we observed that the older subgroups identified based on G1 topography also exhibited the youth-like vs. aged pattern in G2 topography. This pattern was, however, evident also in G3 only in the main sample, possibly suggesting a limited contribution of G3 topography in determining overall functional profiles in older age. In terms of the behavioral relevance of maintaining youth-like gradient topography in older age, we observed effects on word recall performance in both samples; although the Reviewer correctly points out that, the difference between subgroups was significant at trend-level (*p* = 0.058) in the replication dataset. While this indeed underscores the importance of replication efforts in additional samples, we argue that the pattern observed in our replication dataset is overall consistent with, and conveys effects in the expected direction based on, the original observations in our main dataset.

In revising the manuscript, we have performed additional analyses for replication purposes in terms of memory. Originally, we observed a significant association between G2 topography and episodic memory across the main sample. However, this effect did not remain significant after FDR adjustment for multiple comparisons. To evaluate this association further, we conducted a corresponding hierarchical multiple regression analysis in the replication dataset, which supported a role of G2 in memory (Adj. R^2^ = 0.368, ΔR^2^ = 0.081, F = 1.992, *p* = 0.028). Together, these analyses suggest that inter-individual differences in episodic memory performance may in part be explained by the spatial characteristics of G2 across the adult lifespan, although increased statistical power in relation to the large number of TSM parameters included in the hierarchical regression models may be needed to explore this association in smaller, age-stratified, groups. Relatedly, it is worth mentioning that higher levels of memory performance in older age were linked to the maintenance of youth-like G2 topography in both our main and replication datasets.

In parallel, topographic parameters of G1 predicted memory performance in the younger adults, which successfully replicates TSM-based results previously reported in Przeździk et al., 2019. Although similar associations were not evident within the other age groups, a link between G1 topography and memory was demonstrated in older age based on (a) the identification of individuals maintaining a youth-like G1 profile and higher levels of memory, within which (b) memory performance was, as in young adults, significantly predicted by G1 topography.

The spatial correspondence between G2 topography and distribution of hippocampal D1DRs was lateralized to the right, and as the Reviewer points out, as such did not replicate across hemispheres. To which extent replication across hemispheres should be expected in this case is, however, difficult to determine. Lateralization and/or hemispheric asymmetry is commonly observed in numerous hippocampal features, from the molecular level to its functional involvement in behavior (Nematis et al., 2023; Persson & Söderlund, 2015), including various dopaminergic markers tested in the animal literature (Afonso et al., 1993; Sadeghi et al., 2017). Yet, potential differences between hemispheres in D1DR availability and the spatial distribution of receptors along hippocampal axes remain less studied in humans. More data is therefore needed to determine the nature of this right-hemisphere lateralization.

In sum, we argue that our results show a good level of replication across independent datasets and across analyses in our main dataset. Whereas this study did not attempt replication of all analyses conducted in the main dataset, it has through replication across independent samples provided support for its main findings – the organization of hippocampal-neocortical connectivity along three main hippocampal gradients across the adult lifespan, and the gradient topography-based identification of older individuals maintaining a youth-like hippocampal organization in older age.

The revised manuscript includes edits made to incorporate the new analyses and clarifications of observations in relation to memory.

In the Results, page 17:

“Observing that the association between G2 and memory did not remain significant after FDR adjustment, we performed the same analysis in our replication dataset, which also included episodic memory testing. Consistent with the observation in our main dataset, G2 significantly predicted memory performance (Adj. R^2^ = 0.368, ΔR^2^ = 0.081, F = 1.992, *p* = 0.028) over and above covariates and topography of G1. Here, the analysis also showed that G1 topography predicted performance across the sample (Adj. R^2^ = 0.325, ΔR^2^ = 0.112, F = 3.431, *p* < 0.001).”

In the Discussion, page 26:

“Results linked both G1 and G2 to episodic memory, suggesting complimentary contributions of these two overlapping long-axis modes. Considered together, analyses in the main and replication datasets indicated a role of G2 topography in memory across the adult lifespan, independent of age. A similar association with G1 was only evident across the entire sample in the replication dataset, whereas results in the main sample seemed to emphasize a role of youthlike G1 topography in memory performance. In line with previous research, memory was successfully predicted by G1 topography in young adults(30), and similarly predicted by G1 in older adults exhibiting a youth-like functional profile.”

(4) Please share the data and code and add a description of data and code availability in the manuscript.

We have now made our code available, and added a statement on data and code availability in the revised manuscript.

On page 37: “Data from the DyNAMiC study are not publicly available. Access to the original data may be shared upon request from the Principal investigator, Dr. Alireza Salami. The Matlab, R, and FSL codes used for analyses included in this study are openly available at https://github.com/kristinnordin/hcgradients. Computation of gradients was done using the freely available toolbox ConGrads: https://github.com/koenhaak/congrads.”

**Reviewer #3 (Recommendations For The Authors):**
Please see the comments in the public review.

We thank the Reviewer for their comments and recommendations, and have addressed them in the “Public review” section.

References

Afonso, D., Santana, C., & Rodriguez, M. (1993). Neonatal lateralization of behavior and brain dopaminergic asymmetry. *Brain Research Bulletin*, *32*(1), 11–16. https://doi.org/10.1016/0361-9230(93)90312-Y

DeKraker, J., Haast, R. A., Yousif, M. D., Karat, B., Lau, J. C., Köhler, S., & Khan, A. R. (2022). Automated hippocampal unfolding for morphometry and subfield segmentation with HippUnfold. *eLife*, *11*, e77945. https://doi.org/10.7554/eLife.77945

Dubovyk, V., & Manahan-Vaughan, D. (2019). Gradient of expression of dopamine D2 receptors along the dorso-ventral axis of the hippocampus. *Frontiers in Synaptic Neuroscience*, *11*. https://doi.org/10.3389/fnsyn.2019.00028

Edelmann, E., & Lessmann, V. (2018). Dopaminergic innervation and modulation of hippocampal networks. *Cell and Tissue Research*, *373*(3), 711–727. https://doi.org/10.1007/s00441-018-2800-7

Gasbarri, A., Verney, C., Innocenzi, R., Campana, E., & Pacitti, C. (1994). Mesolimbic dopaminergic neurons innervating the hippocampal formation in the rat: A combined retrograde tracing and immunohistochemical study. *Brain Research*, *668*(1), 71–79. https://doi.org/10.1016/0006-8993(94)90512-6

Glasser, M. F., & Essen, D. C. V. (2011). Mapping Human Cortical Areas In Vivo Based on Myelin Content as Revealed by T1- and T2-Weighted MRI. *Journal of Neuroscience*, *31*(32), 11597–11616. https://doi.org/10.1523/JNEUROSCI.2180-11.2011

Kaller, S., Rullmann, M., Patt, M., Becker, G.-A., Luthardt, J., Girbardt, J., Meyer, P. M., Werner, P., Barthel, H., Bresch, A., Fritz, T. H., Hesse, S., & Sabri, O. (2017). Test– retest measurements of dopamine D1-type receptors using simultaneous PET/MRI imaging. *European Journal of Nuclear Medicine and Molecular Imaging*, *44*(6), 1025–1032. https://doi.org/10.1007/s00259-017-3645-0

Katsumi, Y., Zhang, J., Chen, D., Kamona, N., Bunce, J. G., Hutchinson, J. B., Yarossi, M., Tunik, E., Dickerson, B. C., Quigley, K. S., & Barrett, L. F. (2023). Correspondence of functional connectivity gradients across human isocortex, cerebellum, and hippocampus. *Communications Biology*, *6*(1), Article 1. https://doi.org/10.1038/s42003-023-04796-0

Kempadoo, K. A., Mosharov, E. V., Choi, S. J., Sulzer, D., & Kandel, E. R. (2016). Dopamine release from the locus coeruleus to the dorsal hippocampus promotes spatial learning and memory. *Proceedings of the National Academy of Sciences*, *113*(51), 14835–14840. https://doi.org/10.1073/pnas.1616515114

Navarro Schröder, T., Haak, K. V., Zaragoza Jimenez, N. I., Beckmann, C. F., & Doeller, C. F. (2015). Functional topography of the human entorhinal cortex. *eLife*, *4*, e06738. https://doi.org/10.7554/eLife.06738

Nemati, S. S., Sadeghi, L., Dehghan, G., & Sheibani, N. (2023). Lateralization of the hippocampus: A review of molecular, functional, and physiological properties in health and disease. *Behavioural Brain Research*, *454*, 114657. https://doi.org/10.1016/j.bbr.2023.114657

Nilsson, L.-G., Adolfsson, R., Bäckman, L., Frias, C. M. de, Molander, B., & Nyberg, L. (2004). Betula: A Prospective Cohort Study on Memory, Health and Aging. *Aging, Neuropsychology, and Cognition*, *11*(2–3), 134–148. https://doi.org/10.1080/13825580490511026

Nyberg, L. (2017). Functional brain imaging of episodic memory decline in ageing. *Journal of Internal Medicine*, *281*(1), 65–74. https://doi.org/10.1111/joim.12533

Nyberg, L., Boraxbekk, C.-J., Sörman, D. E., Hansson, P., Herlitz, A., Kauppi, K., Ljungberg, J. K., Lövheim, H., Lundquist, A., Adolfsson, A. N., Oudin, A., Pudas, S., Rönnlund, M., Stiernstedt, M., Sundström, A., & Adolfsson, R. (2020). Biological and environmental predictors of heterogeneity in neurocognitive ageing: Evidence from Betula and other longitudinal studies. *Ageing Research Reviews*, *64*, 101184. https://doi.org/10.1016/j.arr.2020.101184

Paquola, C., Benkarim, O., DeKraker, J., Larivière, S., Frässle, S., Royer, J., Tavakol, S., Valk, S., Bernasconi, A., Bernasconi, N., Khan, A., Evans, A. C., Razi, A., Smallwood, J., & Bernhardt, B. C. (2020). Convergence of cortical types and functional motifs in the human mesiotemporal lobe. *eLife*, *9*, e60673. https://doi.org/10.7554/eLife.60673

Pedersen, R., Johansson, J., Nordin, K., Rieckmann, A., Wåhlin, A., Nyberg, L., Bäckman, L., & Salami, A. (2024). Dopamine D1-Receptor Organization Contributes to Functional Brain Architecture. *Journal of Neuroscience*, *44*(11). https://doi.org/10.1523/JNEUROSCI.0621-23.2024

Pedersen, R., Johansson, J., & Salami, A. (2023). Dopamine D1-signaling modulates maintenance of functional network segregation in aging. *Aging Brain*, *3*, 100079. https://doi.org/10.1016/j.nbas.2023.100079

Persson, J., & Söderlund, H. (2015). Hippocampal hemispheric and long-axis differentiation of stimulus content during episodic memory encoding and retrieval: An activation likelihood estimation meta-analysis. *Hippocampus*, *25*(12), 1614–1631. https://doi.org/10.1002/hipo.22482

Przeździk, I., Faber, M., Fernández, G., Beckmann, C. F., & Haak, K. V. (2019). The functional organisation of the hippocampus along its long axis is gradual and predicts recollection. *Cortex*, *119*, 324–335. https://doi.org/10.1016/j.cortex.2019.04.015

Sadeghi, L., Rizvanov, A. A., Salafutdinov, I. I., Dabirmanesh, B., Sayyah, M., Fathollahi, Y., & Khajeh, K. (2017). Hippocampal asymmetry: Differences in the left and right hippocampus proteome in the rat model of temporal lobe epilepsy. *Journal of Proteomics*, *154*, 22–29. https://doi.org/10.1016/j.jprot.2016.11.023

Tian, Y., Margulies, D. S., Breakspear, M., & Zalesky, A. (2020). Topographic organization of the human subcortex unveiled with functional connectivity gradients. *Nature Neuroscience*, 1–12. https://doi.org/10.1038/s41593-020-00711-6

vos de Wael, R., Larivière, S., Caldairou, B., Hong, S.-J., Margulies, D. S., Jefferies, E., Bernasconi, A., Smallwood, J., Bernasconi, N., & Bernhardt, B. C. (2018). Anatomical and microstructural determinants of hippocampal subfield functional connectome embedding. *Proceedings of the National Academy of Sciences*, *115*(40), 10154–10159. https://doi.org/10.1073/pnas.1803667115